# Endoplasmic reticulum stress activates human IRE1α through reversible assembly of inactive dimers into small oligomers

Vladislav Belyy[1], Iratxe Zuazo-Gaztelu[2], Andrew Alamban[1], Avi Ashkenazi[2]*, Peter Walter[1,3]*

[1]Department of Biochemistry and Biophysics, University of California, San Francisco, San Francisco, United States; [2]Cancer Immunology, Genentech, Inc, South San Francisco, United States; [3]Howard Hughes Medical Institute, University of California, San Francisco, San Francisco, United States

**Abstract** Protein folding homeostasis in the endoplasmic reticulum (ER) is regulated by a signaling network, termed the unfolded protein response (UPR). Inositol-requiring enzyme 1 (IRE1) is an ER membrane-resident kinase/RNase that mediates signal transmission in the most evolutionarily conserved branch of the UPR. Dimerization and/or higher-order oligomerization of IRE1 are thought to be important for its activation mechanism, yet the actual oligomeric states of inactive, active, and attenuated mammalian IRE1 complexes remain unknown. We developed an automated two-color single-molecule tracking approach to dissect the oligomerization of tagged endogenous human IRE1 in live cells. In contrast to previous models, our data indicate that IRE1 exists as a constitutive homodimer at baseline and assembles into small oligomers upon ER stress. We demonstrate that the formation of inactive dimers and stress-dependent oligomers is fully governed by IRE1's lumenal domain. Phosphorylation of IRE1's kinase domain occurs more slowly than oligomerization and is retained after oligomers disassemble back into dimers. Our findings suggest that assembly of IRE1 dimers into larger oligomers specifically enables *trans*-autophosphorylation, which in turn drives IRE1's RNase activity.

*For correspondence:
aa@gene.com (AA);
pwalter@altoslabs.com (PW)

## Editor's evaluation

In this study, Belyy et al., have developed a powerful new imaging system--one that will benefit others in the cell biology community--to measure how a key transducer of the unfolded protein response , Ire1, responds to endoplasmic reticulum (ER) stress. While prior studies indicated the existence of large Ire1 oligomers that arose in the ER membrane after stress, this study used single molecule tracking and native levels of Ire1 to demonstrate that Ire1 naturally exists in the inactive state as a dimer and that much smaller oligomers form after stress, a phenomenon governed by the Ire1 lumenal domain. Moreover, Ire1 trans-phosphorylation, which is required for activation, begins after oligomer formation. Overall, these studies yield unprecedented insight into the mechanism underlying the unfolded protein response and have revised our understanding of Ire1 dynamics.

## Introduction

Protein oligomerization is central to cell biology. The regulated assembly of membrane proteins into dimers or larger oligomers constitutes a fundamental cellular mechanism for relaying information across membranes. The majority of receptor superfamilies rely on some form of oligomerization, including G-protein coupled receptors (*Gurevich and Gurevich, 2018*), integrins (*Shattil and*

**eLife digest** Our cells contain many different compartments that each perform specific tasks. A cellular compartment known as the endoplasmic reticulum is responsible for making many of the proteins the cell requires and transporting them around the cell.

It is important that the endoplasmic reticulum remains healthy and, therefore, cells use a protein called IRE1 that senses when this compartment is under stress. IRE1 then sends a signal to the control center of the cell (known as the nucleus) to ask for help. Previous studies suggest that IRE1 assembles into either pairs or larger groups of molecules known as oligomers when it senses that the endoplasmic reticulum is under stress. However, it remains unclear whether such assembly is the main switch that turns IRE1 on and, if so, how many molecules need to come together to flip the switch.

Here, Belyy et al. genetically engineered human bone cancer cells to attach a mark known as a HaloTag to IRE1.The team developed a microscopy approach to count, in living cells, how many tagged IRE1 molecules join. The experiments indicated that IRE1 proteins were generally found as pairs in unstressed cells. When the endoplasmic reticulum experienced stress, IRE1 proteins briefly assembled into oligomers before disassembling back into pairs. Mutated versions of IRE1 revealed the exact parts of IRE1 that connect the pairs and the larger oligomers.

These findings suggest that the assembly of IRE1 pairs into oligomers plays a major part in the activation of IRE1 to send a stress signal to the nucleus. IRE1 signaling is closely implicated in both cancer biology and aging, and therefore, understanding how it works may aid the development of new therapies for cancer, dementia, and other health conditions affecting older people. Furthermore, the microscopy approach developed in this work could be adapted to study other proteins that relay signals in living cells.

*Newman, 2004*), receptor tyrosine kinases (*Chung, 2017*), T-cell receptors (*Reich et al., 1997*), and death receptors (*Ashkenazi and Dixit, 1998*). While significant progress has been made in understanding oligomeric assembly of cell-surface receptors, much less is known about oligomerization of intracellular membrane proteins. This is in part because intracellular oligomers are often too small, dynamic, or weakly associated to be resolved by conventional approaches.

One such oligomer-forming protein is the ER membrane-resident stress sensor IRE1. It is a dual-function kinase/ribonuclease (RNase) responsible for initiating the most evolutionarily conserved branch of the unfolded protein response (UPR) (*Kaufman, 1999*; *Cox et al., 1993*; *Mori et al., 1993*). The UPR is a major signaling network that lies at the core of cellular homeostasis and is responsible for making cellular life-or-death decisions when faced with an imbalance between protein folding load and capacity of the ER's protein folding machinery (*Ron and Walter, 2007*; *Walter and Ron, 2011*). IRE1's role as a master regulator of the UPR has made it an important subject of both basic and translational investigation. Upon its activation by the buildup of unfolded proteins in the ER lumen, IRE1 undergoes kinase-mediated *trans*-autophosphorylation and catalyzes RNase-mediated non-conventional splicing of the *XBP1* mRNA (*Tirasophon et al., 1998*; *Yoshida et al., 2001*) (in mammals; *HAC1* mRNA in yeast) (*Sidrauski and Walter, 1997*) as well as the decay of multiple mRNA targets via processes termed regulated IRE1-decay (RIDD) (*Hollien and Weissman, 2006*; *Moore and Hollien, 2015*; *Hollien et al., 2009*) and RIDD lacking endomotif (RIDDLE) (*Le Thomas et al., 2021*). While IRE1's activation is generally thought to involve the formation of dimers and/or larger oligomers, the extent and functional importance of this oligomerization phenomenon, along with the precise oligomeric state of IRE1 complexes, remain hotly debated.

Early work on yeast IRE1 revealed that both the lumenal (*Credle et al., 2005*) and cytosolic (*Korennykh et al., 2009*) domains can individually crystallize as helical filaments, and that IRE1 molecules assemble into puncta in the ER membrane upon induction of ER stress (*Kimata et al., 2007*). Similarly, fluorescently tagged human IRE1α ('IRE1' hereafter) was observed to reversibly assemble into large, topologically complex puncta in a stress-dependent fashion (*Li et al., 2010*; *Belyy et al., 2020*; *Tran et al., 2021*). The Hill coefficients for purified yeast (*Korennykh et al., 2009*) and human (*Li et al., 2010*) IRE1 kinase/RNase domains were measured to be ~8 and ~ 3.4, respectively, indicating that the cooperative formation of oligomers larger than dimers plays an important role in IRE1's enzymatic cycle. The lumenal domain, while itself lacking catalytic activity, was also observed to assemble into

dimers and larger oligomers in vitro (*Gardner and Walter, 2011*; *Karagöz et al., 2017*). This assembly occurs across two predicted interfaces: IF1$^L$ (L for lumenal), generally accepted to be the primary dimerization interface, and IF2$^L$ (*Figure 1A*), which mediates higher order oligomerization.

Despite this wealth of information, the oligomeric state of both active and inactive IRE1 complexes in mammalian cells remains unclear. It has been alternatively proposed that the monomer-to-dimer transition serves as the main activation signal and that the formation of high-order oligomers is instead the primary regulatory step. The former is supported by the observation of stress-induced increase in crosslinking of a Q105C mutant engineered into the IF1$^L$ interface (*Amin-Wetzel et al., 2017*), while the latter rests on the observation of large clusters of fluorescently tagged IRE1 in stressed cells and on the finding that genetic disruption of the IF2$^L$ interface abrogates IRE1 activity (*Karagöz et al., 2017*). However, crosslinking of a single residue is not necessarily proportional to the degree of dimerization. Indeed, the dimer of IRE1's lumenal domains has been predicted to undergo substantial conformation changes upon peptide binding (*Karagöz et al., 2017*), which alongside the biochemical changes in the lumen of an acutely stressed ER may alter crosslinking efficiency. Most other studies relied on exogenous overexpression of tagged IRE1, which may in turn bias the equilibrium of an oligomerization-prone protein away from physiologically relevant levels. To pursue an orthogonal strategy, we set out to directly measure the oligomerization of endogenously labeled IRE1 in live human cells. To this end, we developed a single-molecule microscopy approach that proved useful to reveal the precise oligomeric changes that underpin IRE1 activation. More broadly, this approach promises to provide a powerful tool to study the oligomerization of other proteins residing on internal membranes in eukaryotic cells.

## Results

### Endogenously tagged IRE1 Is fully active despite not forming large clusters

To study the oligomerization of endogenous IRE1, we inserted a C-terminal HaloTag (*Los et al., 2008*) into IRE1's genomic locus in U-2 OS cells using CRISPR/Cas9-based gene editing (*Figure 1A*). Following clonal selection, we chose a clone that satisfied the following criteria: 1) comparable IRE1 expression levels to unedited U-2 OS cells, 2) absence of wild-type (WT) IRE1 protein lacking the HaloTag, and 3) intact UPR activation in response to ER stress. Additionally, we selected a second clone with lower levels of IRE1-HaloTag protein to evaluate effects of expression level and rule out clonal artifacts (*Figure 1—figure supplement 1*). Comparing expression of IRE1-HaloTag in the two clones to endogenous IRE1 levels by gel band densitometry, we found that the higher-expressing clone has ~3.5-fold higher levels of IRE1-HaloTag protein than WT (averaged across stress conditions) while the lower-expressing clone contains levels of IRE1-HaloTag ~ 30% lower than WT. UPR activation in IRE1-HaloTag clones was ascertained by the detection of ER stress-dependent *XBP1* mRNA splicing (*Figure 1B* and *Figure 1—figure supplement 2*), IRE1 phosphorylation, production of XBP1s protein, upregulation of the CHOP and ATF4 transcription factors, and cleavage of ATF6 (*Figure 1C* and *Figure 1—figure supplement 1*). Furthermore, we found that IRE1-HaloTag cells exhibited RIDD activity, as demonstrated by the decay of the previously described RIDD targets DGAT2, BCAM, and TGOLN2 (*Le Thomas et al., 2021*; *Figure 1—figure supplement 2*).We concluded that the endogenous C-terminal HaloTag does not substantially interfere with IRE1's kinase or RNase activity and could provide an excellent way to image IRE1 dynamics in live cells.

A key advantage of HaloTag fusion proteins stems from the fact that they can be labeled with bright and photostable cell-permeable dyes. Thus, despite IRE1 being a comparatively low-abundance protein, we could readily image it by spinning-disk confocal microscopy after labeling it with the JF549 dye conjugated with the HaloTag ligand (*Grimm et al., 2017*). As expected, IRE1-HaloTag exhibited a reticulated distribution characteristic of ER-localized proteins (*Figure 1D*). We were surprised to observe that IRE1-HaloTag did not assemble into large clusters upon induction of ER stress (*Figure 1E*; also see "Detection of Large IRE1 Clusters" in Materials and Methods), in direct contrast to previous work by us and others that relied on ectopic expression of GFP-tagged IRE1 protein (*Li et al., 2010*; *Belyy et al., 2020*; *Ricci et al., 2019*; *Ricci et al., 2021*; *Cohen et al., 2017*; *Wu et al., 2021*; *Li et al., 2020*). The lack of clustering was not due to a defect of the IRE1-HaloTag fusion construct, since overexpression of the same IRE1-HaloTag protein by transient transfection resulted in readily

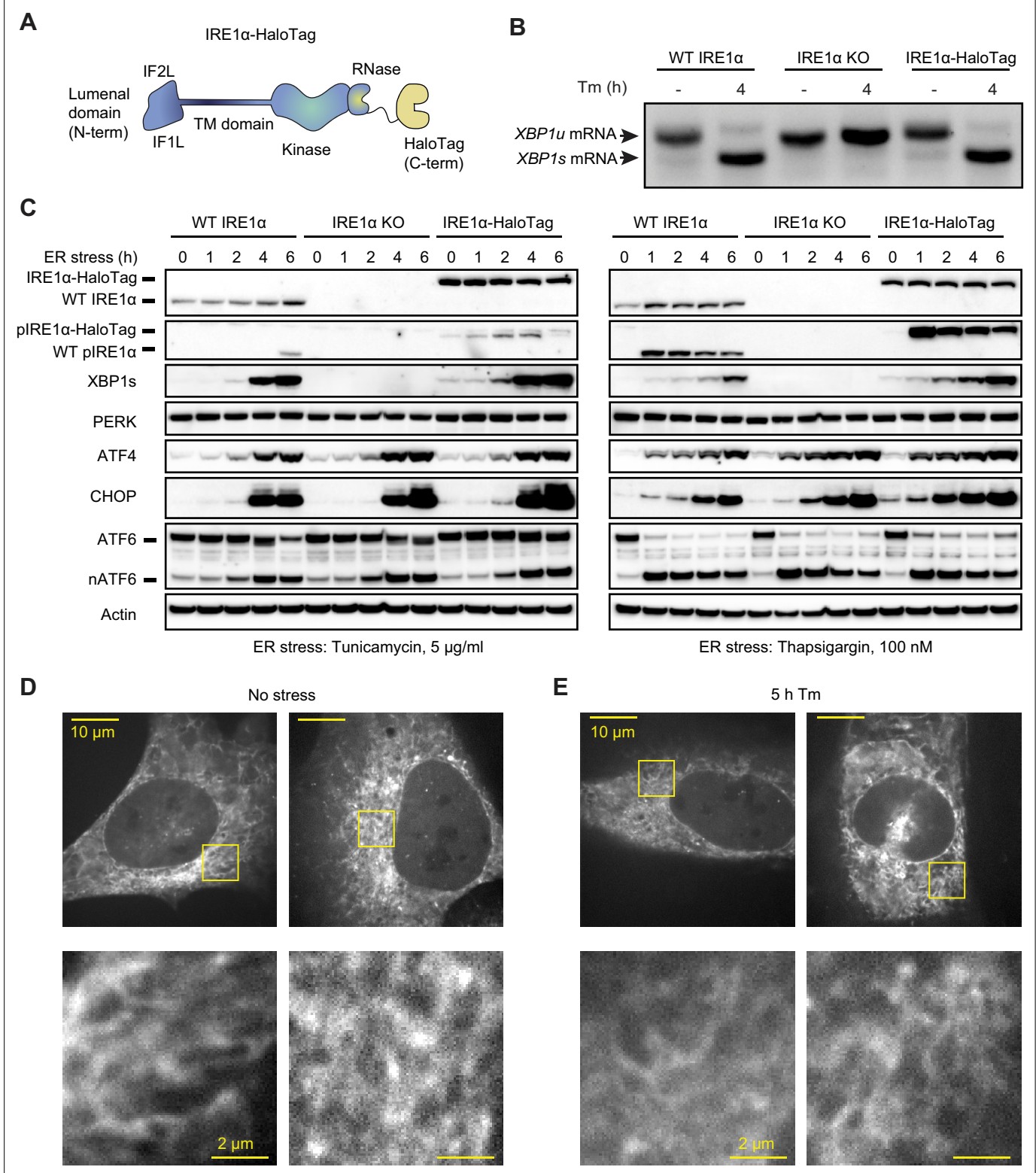

**Figure 1.** Endogenously tagged IRE1α is fully active despite not forming large clusters. (**A**) Schematic representation of IRE1 with a C-terminal HaloTag, the construct used for tagging IRE1 at the endogenous locus. IF1[L] and IF2[L] refer to the primary dimerization and oligomerization interfaces of the lumenal domain, respectively. (**B**) RT-PCR analysis of stress-dependent XBP1 mRNA splicing in WT U-2 OS cells, IRE1 knock-out (KO) U-2 OS cells, and U-2 OS cells in which IRE1 has been fully edited with a C-terminal HaloTag. Tm indicates treatment with 5 µg/ml tunicamycin. (**C**) Immunoblot of UPR activation in response to 5 µg /ml tunicamycin (left) and 100 nM thapsigargin (right) treatments in the three cell lines shown in panel B. (**D**) Maximum

*Figure 1 continued on next page*

*Figure 1 continued*

intensity projections of representative spinning-disk confocal images of live cells expressing endogenously tagged IRE1-HaloTag, labeled with the JF549 dye. Regions shown with yellow boxes are enlarged below. (**E**) Same as D, except the cells have been treated with 5 μg/ml tunicamycin for 5 hr.

The online version of this article includes the following source data and figure supplement(s) for figure 1:

**Source data 1.** Annotated uncropped gel used to generate *Figure 1B*.

**Source data 2.** Raw uncropped gel used to generate *Figure 1B*.

**Source data 3.** All annotated uncropped gels used to generate *Figure 1C*.

**Source data 4.** Raw uncropped gel of immunoblot against IRE1 and phospho-IRE1 in *Figure 1C*.

**Source data 5.** Raw uncropped gel of immunoblot against XBP1 in *Figure 1C*.

**Source data 6.** Raw uncropped gel of immunoblot against PERK and actin in *Figure 1C*.

**Source data 7.** Raw uncropped gel of immunoblot against ATF4 in *Figure 1C*.

**Source data 8.** Raw uncropped gel of immunoblot against ATF6 in *Figure 1C*.

**Source data 9.** Raw uncropped gel of immunoblot against CHOP in *Figure 1C*.

**Figure supplement 1.** Comparison of high- and low-expression clones.

**Figure supplement 1—source data 1.** Annotated uncropped gel used to generate *Figure 1—figure supplement 1A and C*.

**Figure supplement 1—source data 2.** Raw uncropped gel of immunoblot against IRE1 and phospho-IRE1 in *Figure 1—figure supplement 1A*.

**Figure supplement 1—source data 3.** Raw uncropped gel of immunoblot against XBP1 in *Figure 1—figure supplement 1A*.

**Figure supplement 1—source data 4.** Raw uncropped gel of immunoblot against PERK in *Figure 1—figure supplement 1A*.

**Figure supplement 1—source data 5.** Raw uncropped gel of immunoblot against ATF4 and CHOP in *Figure 1—figure supplement 1A*.

**Figure supplement 1—source data 6.** Raw uncropped gel of immunoblot against ATF6 in *Figure 1—figure supplement 1A*.

**Figure supplement 1—source data 7.** Raw uncropped gel of immunoblot against actin in *Figure 1—figure supplement 1A*.

**Figure supplement 2.** qPCR analysis of of XBP1 splicing, RIDD, and RIDDLE activity in IRE1-HaloTag cells.

**Figure supplement 3.** Examples of stress-induced clustering of IRE1-HaloTag in the context of overexpression.

observed stress-induced clusters (*Figure 1—figure supplement 3*). While unexpected, this observation does not rule out lower-order IRE1 oligomerization at endogenous expression levels, since the limited sensitivity of confocal microscopy would preclude the detection of small oligomers such as dimers or tetramers as distinct morphological features. We therefore sought to devise a more sensitive approach for detecting small oligomers in the ER membrane.

## Development of a two-color tracking algorithm for the detection of small oligomers

Detection of small protein oligomers inside intact cells is a notoriously challenging task. A range of approaches, each carrying a unique set of strengths and limitations, has been employed in the past (*Berggård et al., 2007*; *Shashkova and Leake, 2017*; *Sekar and Periasamy, 2003*; *Gell et al., 2012*). We leveraged the fact that the HaloTag protein can be labeled with cell-permeable fluorophores of different colors.

In principle, if a protein is stochastically labeled by fluorophores with distinct spectra and subsequently imaged with single-molecule resolution, its average oligomeric state can be determined by quantifying the fraction of particles that fluoresce in more than one color. For a diffusing protein in live cells, identification of correlated trajectories over multiple frames can boost the accuracy of the analysis (*Figure 2A, B*). However, to date this approach has been limited to reconstituted in vitro systems and plasma membrane-bound proteins (*Low-Nam et al., 2011*; *Coban et al., 2015*; *Schlager et al., 2014*; *Hänselmann and Herten, 2017*; *Valley et al., 2015*; *Steinkamp et al., 2014*; *Li et al., 2021*; *Stüber et al., 2021*). Furthermore, previous implementations lacked experimental controls of defined stoichiometry, relying on a number of physical assumptions to estimate the degree of oligomerization. To overcome these challenges, we developed a fully automated image analysis pipeline for identification of co-localizing, two-color trajectories of ER membrane-resident proteins.

First, we calibrated our tracking-based approach using ER membrane-tethered proteins with well-defined oligomeric states. We expressed in U-2 OS cells synthetic constructs containing either one, two, or four ER-targeted HaloTag proteins (*Figure 2C* and *Figure 2—figure supplement 1*), under

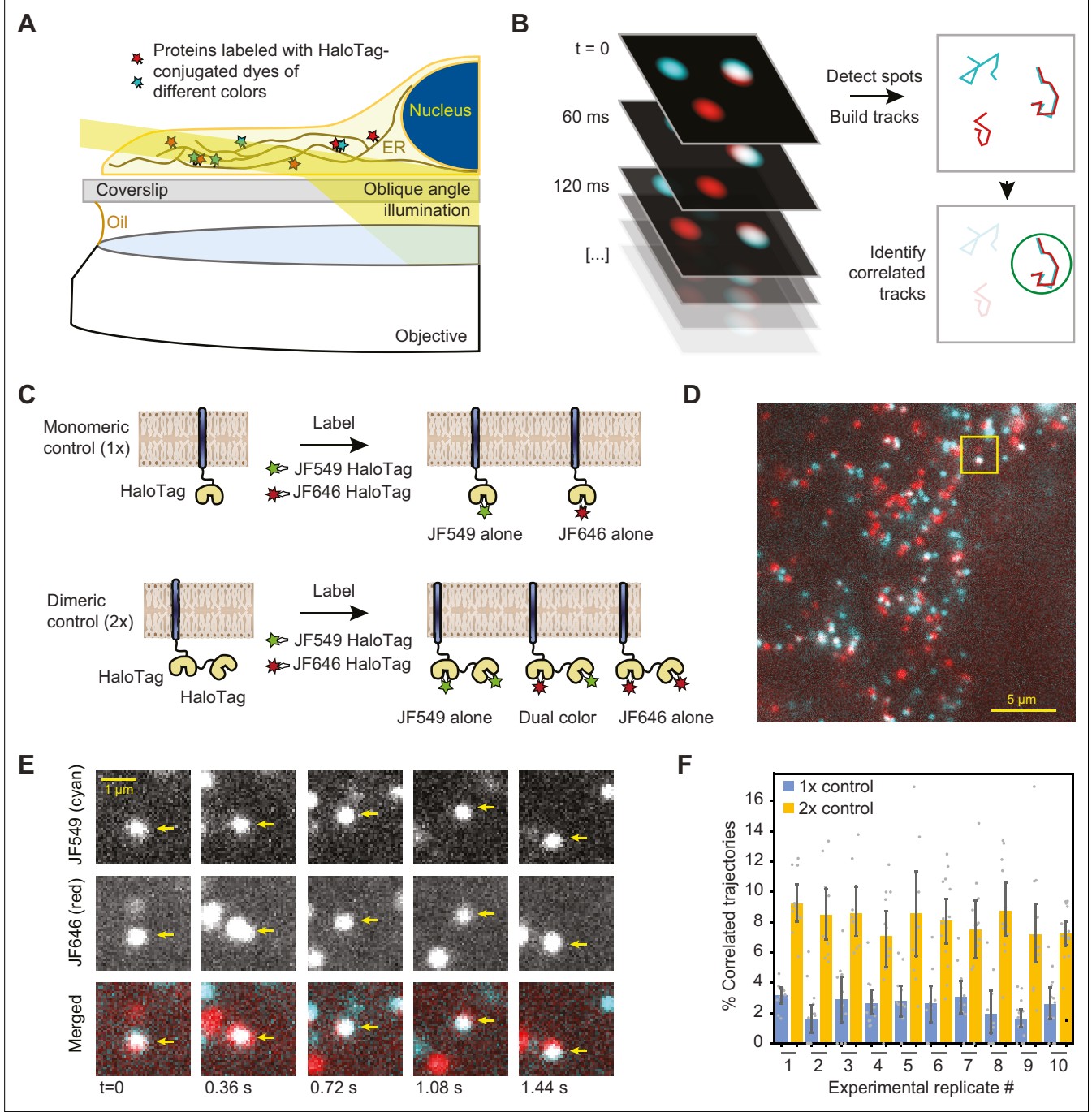

**Figure 2.** Single-particle tracking approach for detection of small oligomers. (**A**) Schematic depiction of the assay. Cells expressing low levels of HaloTag-conjugated proteins are labeled with a mixture of HaloTag-conjugated dyes and imaged by oblique angle illumination. (**B**) Principle behind the analysis of single-particle data. Fluorescent spots are independently tracked in two channels, and correlated trajectories are identified computationally. (**C**) Design of the 1 x and 2 x HaloTag controls. (**D**) Representative frame from a movie of a cell expressing an ER-tethered 2 x tandem HaloTag and labeled with a mixture of JF549 (cyan) and JF646 (red) dyes. (**E**) Several frames of the boxed region in panel D, with co-localizing spots identified with arrows. (**F**) Percentage of correlated trajectories from cells expressing the 1 x and 2 x HaloTag controls, comparing data collected in 10 independent experimental replicates. Each data point represents a single cell, typically comprising several hundred trajectories. Error bars represent 95% confidence intervals.

The online version of this article includes the following source data and figure supplement(s) for figure 2:

**Source data 1.** Pairwise significance test values (permutation test with 10,000 iterations and two-tailed t-test) for conditions plotted in *Figure 2F*.

**Figure supplement 1.** Orthogonal oligomerization controls.

*Figure 2 continued on next page*

*Figure 2 continued*

**Figure supplement 1—source data 1.** Pairwise significance test values (permutation test with 10,000 iterations and two-tailed t-test) for all plotted conditions.

**Figure supplement 2.** Confocal microscopy images of HaloTag controls.

**Figure supplement 3.** Model for fraction of observed % correlated trajectories as a function of true oligomeric state.

control of a weakened CMVd3 promoter (*Slater et al., 2008*). After labeling to saturation with a mixture of JF-549 HaloTag and JF-646 HaloTag dyes, we imaged the cells by oblique angle illumination microscopy. In longer-exposure movies, it was apparent that all HaloTag constructs exhibited a reticulated distribution characteristic of the ER; ER localization was also confirmed by confocal microscopy (*Figure 2—figure supplement 2*). The thin, spread-out morphology of U-2 OS cells, together with the exceptional photophysical properties of the JF dyes, allowed us to readily distinguish single diffusing molecules in both channels and track them over multiple frames (*Figure 2D, E*). As expected, a large number of seconds-long correlated two-color trajectories were observed in cells expressing the tandem 2 x HaloTag construct (*Figure 2D*, *Video 1*), but not in cells expressing the single HaloTag construct.

To quantify the fraction of co-localizing spots, we employed the following algorithm. First, spots were automatically detected and tracked in both channels. Then, the tracks were binned into short trajectories using a sliding window of either 12 or 14 frames (0.72 or 0.84 s) to minimize the ambiguity in assignment of crossing tracks. Pearson's correlation coefficients were then calculated between both the x- and y-coordinates of adjacent tracks within each sliding window. Every track in the JF-549 channel that contained at least one window with a correlation coefficient above a predetermined threshold was classified as a co-localizer (see **Materials and methods** for details). By repeating this analysis on data collected from cells expressing the 1 x and 2 x HaloTag controls in ten independent replicates ( > 10 cells and >1500 trajectories per condition in each replicate), we verified that the algorithm robustly and reproducibly distinguishes between monomeric and dimeric molecules in the ER membrane (*Figure 2F*). To rule out the remote possibility that the 2 x tandem HaloTag protein may be an imperfect control due to differential ligand accessibility of the internal and C-terminal HaloTag proteins, we repeated the analysis with a construct that instead relies on GST dimerization to bring two ER-bound C-terminal HaloTag proteins together (*Figure 2—figure supplement 1*). The measured percentage of co-localized trajectories for this construct was statistically indistinguishable (*P* = 0.58, two-tailed T-test) from that of the tandem 2 x HaloTag protein, indicating that both HaloTag binding sites of a tandem construct remain fully accessible to dye molecules. The 4 x HaloTag construct, created by GST-induced dimerization of tandem HaloTag dimers, exhibited a substantial further increase in the percentage of correlated trajectories (*Figure 2—figure supplement 1*). Using a simple combinatoric model relating the observed correlated trajectories to true oligomeric state (See **Materials and methods** for details), we demonstrated that our assay provides excellent sensitivity for oligomers in the monomer to tetramer range, with sensitivity likely decreasing for oligomers comprised of more than ~10 protomers (*Figure 2—figure supplement 3*).

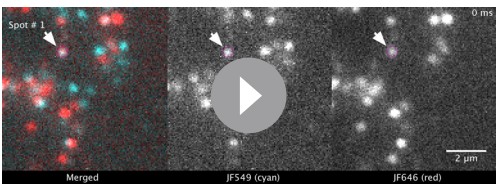

**Video 1.** Co-localizing spots in cells expressing 2 x tandem HaloTag. A cropped and annotated movie recorded from an IRE1 KO cells transiently transfected with the 2 x tandem HaloTag construct and labeled with a mixture of JF549 and JF646 dyes. Two separate co-localizing spots (as determined by the automated analysis pipeline) are annotated.

https://elifesciences.org/articles/74342/figures#video1

## IRE1 transitions from dimers to small oligomers upon ER stress

Having validated the ability to robustly detect changes in oligomeric state, we applied our analysis to cells expressing endogenously tagged IRE1 (*Figure 3A*). We could clearly observe individual fluorescent spots corresponding to single IRE1 molecules moving along ER tubules (*Figure 3B, C*). A fraction of diffusing spots co-localized between the two channels, indicating a significant degree of IRE1 oligomerization even in the absence of stress induction (*Figure 3D, E*). In fact, upon quantification, the fraction of co-localized IRE1 trajectories in non-stressed cells

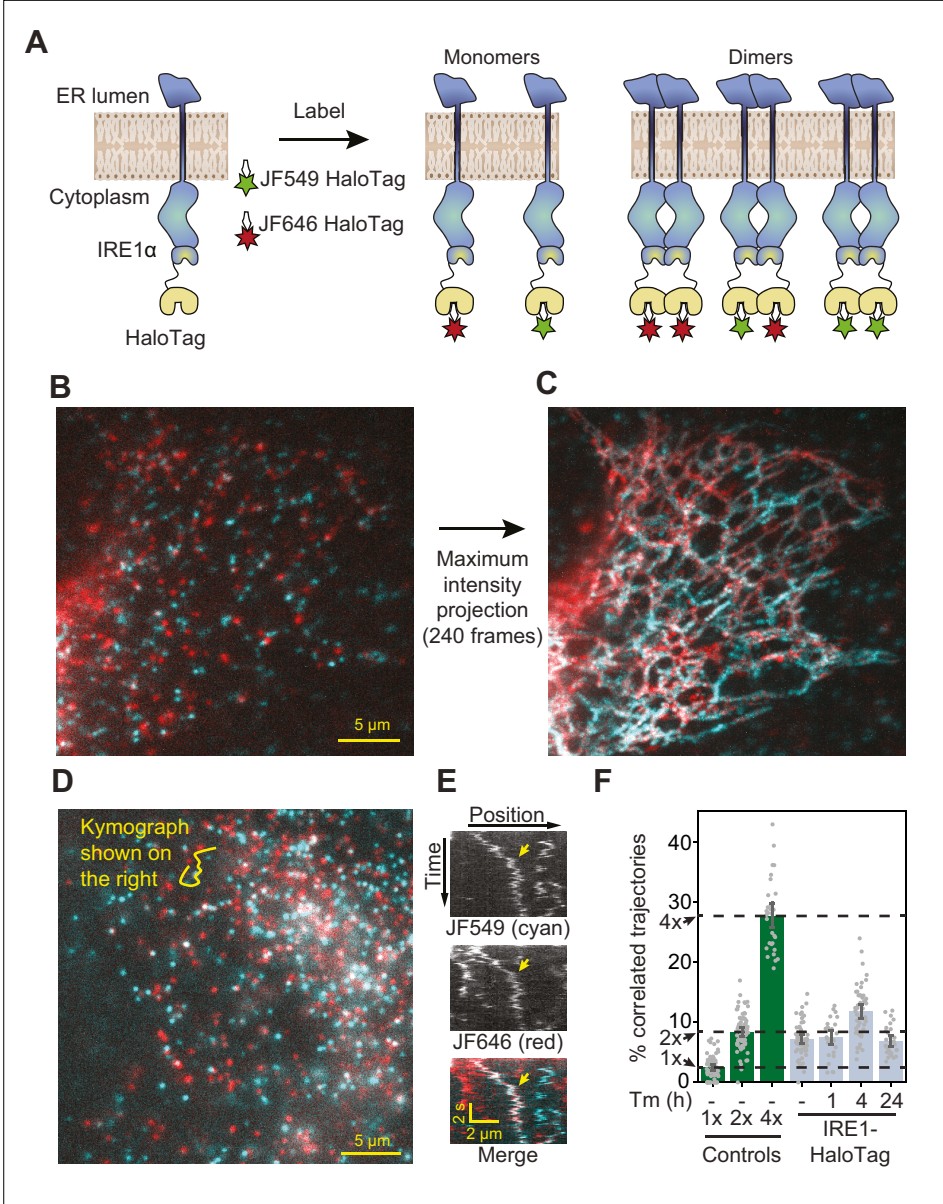

**Figure 3.** Detection of IRE1 dimers and oligomers in live cells. (**A**) Schematic depiction of the assay. IRE1-HaloTag is simultaneously labeled with HaloTag dyes of two different colors, JF549 and JF646. If the protein is purely monomeric, all single-molecule tracks are expected to be either one color or the other. If it is purely dimeric, a fraction of tracks will contain both colors. Such dual-color tracks can then be identified as correlated trajectories. (**B**) Single frame from a long-exposure movie (100ms per frame) of a cell in which IRE1-HaloTag is labeled with a mixture of JF549 (cyan) and JF646 (red) dyes. (**C**) Maximum intensity projection of the entire movie from panel B showing that single IRE1 molecules diffuse along ER tubules. (**D**) Single frame from a short-exposure movie (50ms per frame) of a cell in which IRE1-HaloTag is labeled with a mixture of JF549 (cyan) and JF646 (red) dyes. (**E**) Kymograph (time vs. position plot) along the line shown in panel D. Co-localizing diffusional IRE1 trajectory is shown with a yellow arrow. (**F**) Stress-induced changes in IRE1 oligomerization in response to treatment with 5 µg/ml tunicamycin (Tm), as quantified by the fraction of correlated trajectories. Green bars on the left correspond to the 1 x, 2 x, and 4 x HaloTag controls, respectively. Error bars represent 95% confidence intervals.

The online version of this article includes the following source data and figure supplement(s) for figure 3:

**Source data 1.** Pairwise significance test values (permutation test with 10,000 iterations and two-tailed t-test) for conditions plotted in *Figure 3F*.

**Figure supplement 1.** Effect of ER stress on HaloTag controls.

*Figure 3 continued on next page*

*Figure 3 continued*

**Figure supplement 1—source data 1.** Pairwise significance test values (permutation test with 10,000 iterations and two-tailed t-test) for all plotted conditions.

**Figure supplement 2.** Effect of ER stress on the efficiency of HaloTag labeling.

**Figure supplement 3.** Quantification of diffusion from single-particle trajectories.

**Figure supplement 3—source data 1.** Pairwise significance test values (permutation test with 10,000 iterations and two-tailed t-test) for all plotted conditions.

---

appeared nearly identical to that of the 2 x HaloTag control, strongly suggesting that nearly all IRE1 proteins are pre-assembled into dimers at baseline (*Figure 3F*). Treatment with the glycosylation inhibitor and potent UPR activator tunicamycin (Tm) resulted in a pronounced increase in the fraction of correlated trajectories after 4 hr, indicating that a significant fraction of IRE1 dimers assembled into higher order oligomers. Our model estimates that the mean number of molecules per cluster increases from ~1.8 to ~ 2.5 upon Tm stress (see **Materials and methods** for details). Since our approach does not reveal the individual oligomeric state of any given tracked protein, this observed change is most readily explained by a Tm-dependent shift in equilibrium towards a mixture of dimers and tetramers. Extending the treatment to 24 hr reversed the shift in correlated trajectories, suggesting that IRE1 oligomers dissociate back into dimers under prolonged stress. This finding parallels the previously observed attenuation of IRE1 activity upon prolonged, unmitigated ER stress (*Li et al., 2010*; *Belyy et al., 2020*; *Chang et al., 2018*; *Lin et al., 2007*) (*Figure 3F*).

Because our analysis is rooted in actively identifying correlated diffusive trajectories from single-molecule data, we needed to verify that the induction of ER stress does not substantially alter the efficiency of dye labeling or the diffusion of our proteins of interest. Addition of Tm did not induce an increase in the fraction of correlated trajectories of the 1 x and 2 x HaloTag controls (*Figure 3—figure supplement 1*) and did not affect the efficiency of labeling IRE1-HaloTag with JF dyes (*Figure 3—figure supplement 2*). To examine IRE1 diffusion, we measured apparent diffusion coefficients from single-particle tracks using both the mean-squared displacement (MSD) approach and a recently published state array (SA) method, which is specifically optimized for extracting diffusion coefficients from noisy single-molecule data (*Heckert et al., 2021*). Both approaches showed that diffusion coefficients remain nearly identical among HaloTag controls and IRE1-HaloTag molecules in stressed and unstressed cells (*Figure 3—figure supplement 3*). As expected, the SA method resulted in a far tighter distribution than the MSD method and yielded a mean apparent diffusion coefficient of $0.18 \pm 0.02 \ \mu m^2 \ s^{-1}$ for unstressed IRE1-HaloTag, which is remarkably close to the value of $0.24 \pm 0.02 \ \mu m^2 \ s^{-1}$ that we obtained previously for IRE1-mNeonGreen by FRAP (*Belyy et al., 2020*). Collectively, these data demonstrated that the existence of resting-state dimers and their transient assembly into small oligomers are *bona fide* features of IRE1 signaling rather than consequences of stress-dependent remodeling of the ER membrane.

A key aspect of IRE1 activation is its *trans*-autophosphorylation. Intriguingly, thapsigargin (Tg), which disrupts ER calcium homeostasis by blocking sarco/endoplasmic reticulum $Ca^{2+}$ pumps (*Lu et al., 2014*), induced IRE1 phosphorylation much more rapidly and strongly than Tm, despite leading to similar overall levels of XBP1s production (*Figure 1C*). This observation prompted us to test whether oligomerization is directly proportional to IRE1 phosphorylation by comparing the effects of different ER stressors. Treatment with dithiothreitol (DTT), which causes protein misfolding by reducing disulfide bonds, induced IRE1 oligomerization to the same extent as Tm (*Figure 4*). However, to our surprise, treatment with a high concentration (100 nM) of Tg did not induce a detectable change in oligomeric state either 2 or 4 hr after treatment. Since Tg is a fast-acting stressor compared to Tm, we reasoned that the apparent lack of oligomerization in response to Tg might be explained by a rapid formation and dissolution of IRE1 oligomers, which could be effectively complete by the 2 hr time-point. Indeed, imaging cells only 10 min after the addition of 100 nM Tg revealed a robust increase in IRE1 oligomerization, as indicated by an increase in the fraction of correlated trajectories (*Figure 4*). Furthermore, a lower concentration of 1 nM Tg led to IRE1 oligomerization at the longer 2- and 4 hr time-points. Meanwhile, when 100 nM Tg was combined with saturating Tm, there was no detectable IRE1 oligomerization 4 hr after treatment, demonstrating that the repressive effect of extended Tg treatment overrides the pro-oligomerization effect of Tm. Taken together, our results show that IRE1

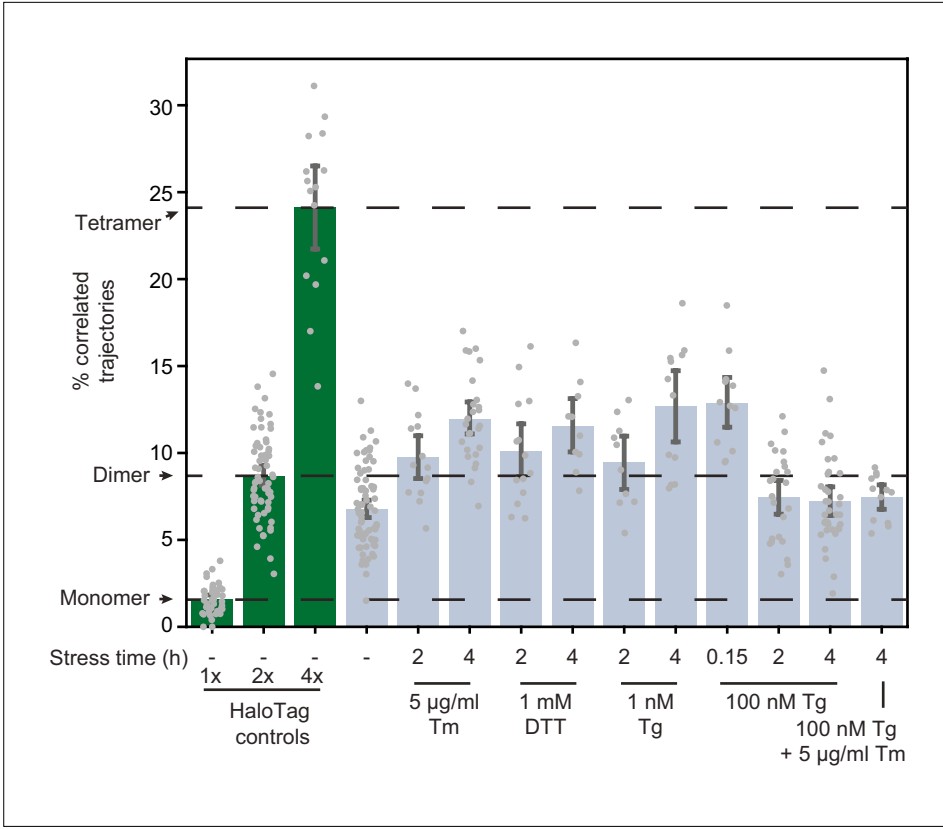

**Figure 4.** Effects of stressors on IRE1 oligomerization. Oligomerization of endogenously tagged IRE1-HaloTag in U-2 OS cells treated with the indicated ER stressors for the indicated amounts of time. Tunicamycin (Tm) inhibits glycosylation in the ER lumen, thapsigargin (Tg) blocks sarco/endoplasmic reticulum $Ca^{2+}$ pumps, and dithiothreitol (DTT) triggers reduction of disulfide bonds. Green bars on the left correspond to the 1 x, 2 x, and 4 x HaloTag controls, respectively. Error bars represent 95% confidence intervals.

The online version of this article includes the following source data and figure supplement(s) for figure 4:

**Source data 1.** Pairwise significance test values (permutation test with 10,000 iterations and two-tailed t-test) for all plotted conditions.

**Figure supplement 1.** Formation of dimers and oligomers in high- and low-expressing clones of IRE1-HaloTag cells.

**Figure supplement 1—source data 1.** Raw uncropped gel of immunoblot against IRE1, XBP1s, and GAPDH of *Figure 4—figure supplement 1*.

**Figure supplement 1—source data 2.** Raw uncropped gel of immunoblot against PERK, ATF4, and CHOP of *Figure 4—figure supplement 1*.

**Figure supplement 1—source data 3.** Raw uncropped gel of immunoblot against ATF6 of *Figure 4—figure supplement 1*.

**Figure supplement 2.** Trajectory density vs. percent correlation.

**Figure supplement 2—source data 1.** Pairwise significance test values (permutation test with 10,000 iterations and two-tailed t-test) for all plotted conditions.

phosphorylation lags behind oligomerization and that all commonly used ER stressors induce IRE1 oligomerization, albeit on different temporal scales. Next, we sought to exclude the formal possibility that the observed dimer-to-oligomer transition was either a clonal artifact or a consequence of differences in expression levels of IRE1 and the control constructs. To this end, we applied our single-particle analysis to the clonal population of cells expressing low levels of IRE1-HaloTag (*Figure 1—figure supplement 1*). IRE1 remained dimeric in unstressed cells even at this decreased expression level (*Figure 4—figure supplement 1*). Meanwhile, the extent of stress-induced oligomerization was

markedly reduced, mirroring the decrease in production of XBP1s protein by these cells (*Figure 1—figure supplement 1*).

To specifically address any potential discrepancies between expression levels of IRE1 and the HaloTag controls, we plotted the measured fraction of correlated trajectories against the total number of trajectories in a given movie. The total number of trajectories served as a proxy for the density of fluorescent spots and, by extension, for the relative abundance of the protein in a cell. We found that protein abundance was comparable for IRE1-HaloTag and the HaloTag controls, and that the differences in the fraction of correlated trajectories remained robustly detectable across a wide range of spot densities (*Figure 4—figure supplement 2*). Finally, to exclude the possibility that labeling with JF dyes could somehow impact the activity of IRE1-HaloTag, we repeated the immunoblot experiment shown in *Figure 1C* under conditions mirroring those used for imaging. We grew cells in clear

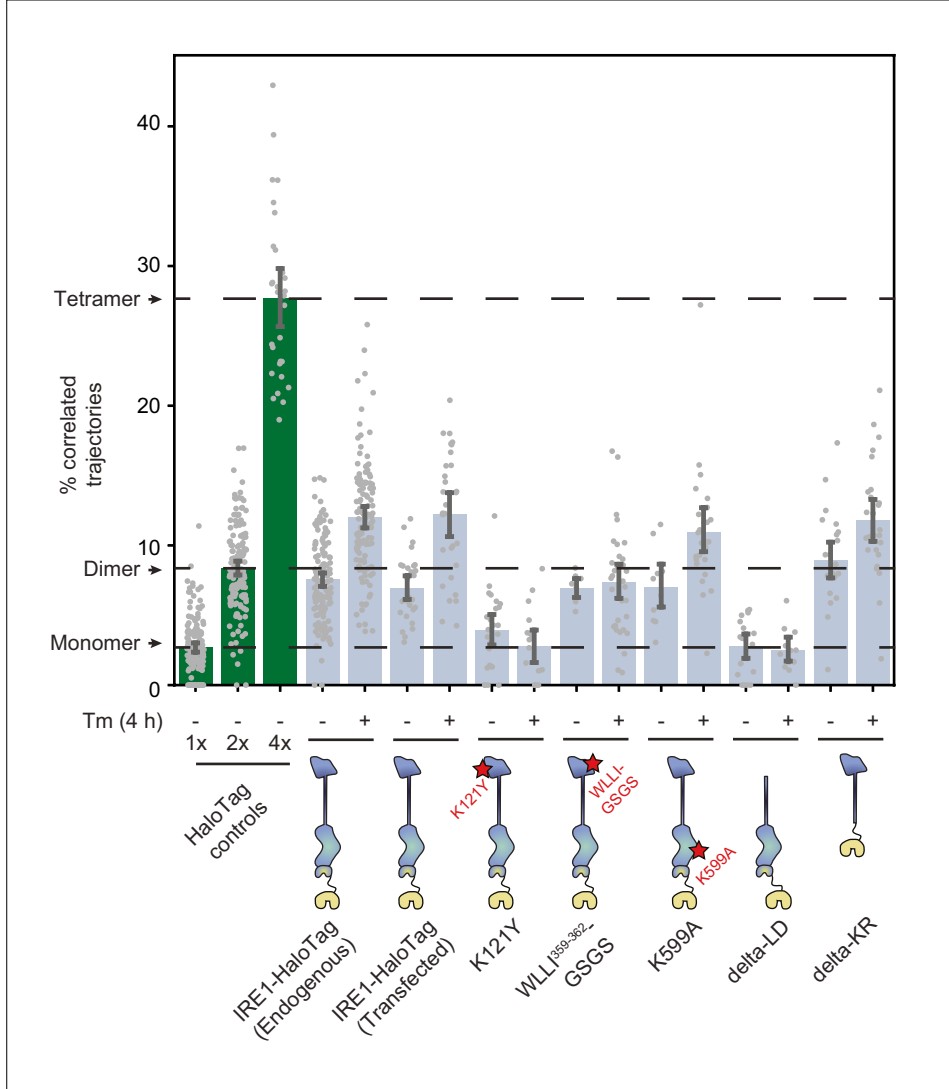

**Figure 5.** Effects of mutations on IRE1 oligomerization. Oligomerization of the indicated IRE1 mutants transiently transfected into IRE1 KO U-2 OS cells and expressed under the control of the weak CMVd3 promoter. 'IRE1-HaloTag (endogenous)' refers to the endogenously tagged IRE1 cells that are shown in *Figure 4*. Error bars represent 95% confidence intervals.

The online version of this article includes the following source data and figure supplement(s) for figure 5:

**Source data 1.** Pairwise significance test values (permutation test with 10,000 iterations and two-tailed t-test) for all plotted conditions.

**Figure supplement 1.** Trajectory counts of all mutants.

FluoroBrite medium on collagen-coated glass sufaces and labeled them with a mixture of JF549 and JF646 dyes exactly as if it were an imaging experiment, finding that activation of IRE1-HaloTag remained intact. We concluded that both the formation of stable IRE1 dimers in non-stressed cells and the assembly of IRE1 into larger oligomers upon stress induction were neither peculiarities of a single clone nor artifacts of expression level or imaging technique, but rather genuine features of IRE1 biology.

## The lumenal domain governs the formation of both dimers and oligomers

To determine which regions of IRE1 are responsible for assembling the protein into dimers and oligomers, we applied our trajectory analysis to a number of key IRE1 mutants (*Figure 5*). Since constructing each mutant by CRISPR technology would have been impractical, we first checked whether the dimer-to-oligomer transition could be measured with transiently transfected IRE1-HaloTag. Indeed, when we expressed IRE1-HaloTag under the control of the truncated CMVd3 promoter in IRE1 KO U-2 OS cells, our method unambiguously detected both the presence of IRE1 dimers in the non-stressed cells and a shift towards larger oligomers upon treatment with Tm. This result paved the way for testing IRE1 mutants in a similar fashion. The first revealing pair of functional mutants comprised K121Y, which disrupts the IF1$^L$ interface of the IRE1 lumenal domain (*Li et al., 2010*), and WLLI$^{359-362}$-GSGS, which disrupts the lumenal domain's oligomerization interface IF2$^L$ (*Karagöz et al., 2017*). These two mutations yielded two starkly different outcomes. The oligomeric state of K121Y remained similar to that of the 1 x HaloTag control, both with and without induction of ER stress. In contrast, WLLI$^{359-362}$-GSGS retained the same oligomeric state as unstressed WT IRE1 both with ($p = 0.74$) and without ($p = 0.18$) ER stress (two-tailed t-test). Thus, both lumenal domain mutants fail to change their oligomeric states in response to ER stress, with K121Y remaining mostly monomeric and WLLI$^{359-362}$-GSGS remaining mostly dimeric.

Next, we probed the potential contribution of the kinase/RNase domain. K599A, a mutation that abrogates IRE1's kinase activity (*Tirasophon et al., 1998*), closely resembled the phenotype of WT IRE1, with only a slightly reduced difference in oligomerization between the non-stress and stress conditions (*Figure 5*). We then tested a pair of more radical mutations: delta-KR, a complete deletion of the kinase/RNase domain, and delta-LD, a complete deletion of the lumenal domain. Remarkably, the delta-LD construct remained purely monomeric regardless of ER stress ($p = 0.92$ and $p = 0.64$ for unstressed and stressed cells, respectively, when compared to the 1 x HaloTag control; two-tailed t-test), while the delta-KR construct recapitulated the stress-dependent transition from dimers to higher order oligomers (*Figure 5*). Particle density analysis confirmed that results from all IRE1 mutants examined are not correlated with the expression levels of the different constructs and represent the mutants' intrinsic propensities for oligomerization (*Figure 5—figure supplement 1*). Taken together, the mutant data confirm the previously proposed role of the lumenal domain as the central governor of IRE1 oligomerization, consistent with its role as the ER stress sensor domain. Both constitutive dimerization and the dimer-to-oligomer transition appear to be entirely controlled by the lumenal domain of IRE1, with the kinase and RNase domains acting downstream.

## Discussion

We measured the stress-dependent oligomeric changes of fully active human IRE1 in live cells at physiological expression levels. Despite IRE1 not forming the massive clusters that were previously observed by us and others in the context of overexpression, we show that the formation of high-order oligomers remains a conserved feature of IRE1's activation. Surprisingly, IRE1 forms constitutive inactive dimers in the absence of externally induced ER stress in U-2 OS cells, thus challenging the widely held notion (*Amin-Wetzel et al., 2017*; *Liu et al., 2000*; *Pincus et al., 2010*) that the monomer-to-dimer equilibrium constitutes the primary regulatory step in IRE1 activation. We demonstrate that the lumenal domain serves as the primary governor of dimerization in the absence of induced stress (via the IF1$^L$ interface) as well as oligomer formation in response to ER stress (via the IF2$^L$ interface). Indeed, the lumenal domain alone is sufficient for the formation of both resting-state dimers and stress-induced oligomers when tethered to the ER membrane, while the kinase/RNase domain alone remains strictly monomeric.

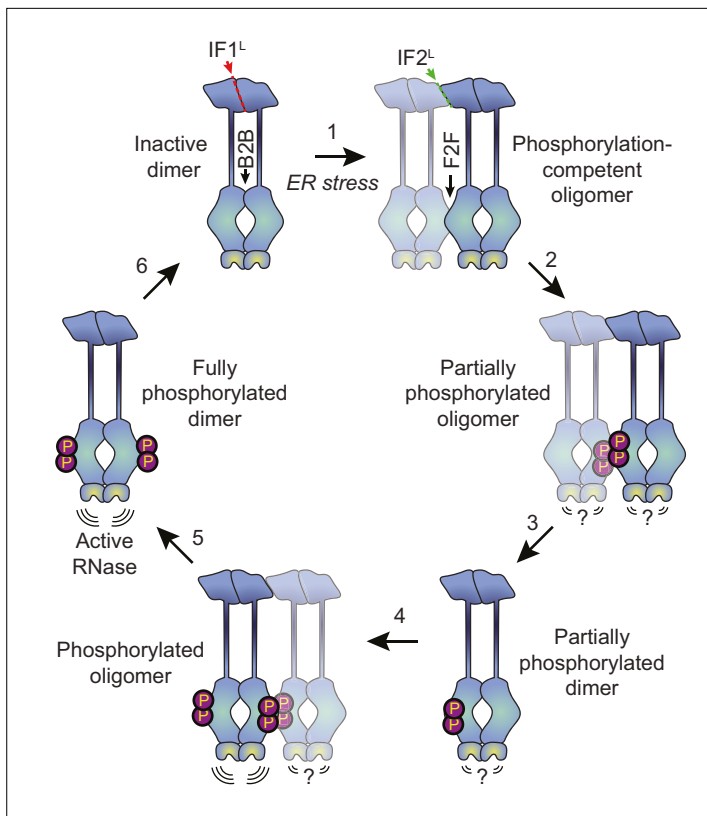

**Figure 6.** Proposed model for human IRE1 activation. In the absence of external stress, IRE1 is pre-assembled into inactive unphosphorylated dimers via the IF1$^L$ interface of the lumenal domain. Kinase domains within the dimer are positioned in a back-to-back (B2B) orientation, which does not allow for phosphorylation. (***Gurevich and Gurevich, 2018***) ER stress forces dimers to oligomerize via the IF2$^L$ interface, placing the kinase active sites of adjacent dimers in a face-to-face (F2F) orientation that favors *trans*-autophosphorylation. Here and throughout the figure, the original dimer is shown in solid blue tones while the newly associated dimer is semi-transparent. (***Shattil and Newman, 2004***) Phosphorylation at the F2F interface results in a partially phosphorylated oligomer, wherein one protomer of each dimer is phosphorylated and one is not. The relative activity of the RNase domains in this state is unknown. (***Chung, 2017***) At this point, the oligomer may dissociate into partially phosphorylated dimers. (***Reich et al., 1997***) Another dimer associates with the partially phosphorylated dimer *via* the second IF2$^L$ interface, catalyzing phosphorylation of the second protomer of the original dimer. Note that this may either occur sequentially, as shown here, or simultaneously with step 2, if multiple dimers assemble into a hexamer or larger oligomer. Phosphorylated IRE1 now has an active RNase domain and dissociates into fully active dimers (***Kaufman, 1999***). Eventually, dimers are dephosphorylated by phosphatases and return back into the inactive state.

Why might the formation of oligomers rather than dimers be the key step in IRE1's activation? The kinase/RNase domains can adopt two distinct dimeric conformations, known as the back-to-back and face-to-face orientations (***Korennykh and Walter, 2012***). The back-to-back orientation is thought to represent the RNase-active form of the protein, but is incapable of trans-autophosphorylation since both kinase active sites face outwards. Meanwhile, the face-to-face arrangement is perfectly suited for trans-autophosphorylation but is not expected to have RNase activity (***Korennykh and Walter, 2012***). It has been proposed that the dimerization of IRE1 allows the kinase/RNase domains to carry out trans-autophosphorylation in a face-to-face orientation, subsequently flipping around to form an active phosphorylated back-to-back dimer (***Ali et al., 2011***). However, this would require a massive rearrangement of cytosolic domains, and the feasibility of such a transition has not been demonstrated. Our data favor an alternative model (***Figure 6***), wherein naive IRE1 is either partially or fully pre-assembled into back-to-back dimers, which remain inactive due to their lack of phosphorylation. In response to ER stress, the lumenal domains drive the assembly of higher order oligomers such as tetramers (that is, dimers of dimers). Since the back-to-back interfaces are already engaged, the formation of tetramers permits inter-dimer face-to-face interactions of kinase-RNase domains, thus

enabling the phosphorylation reaction that in turn activates the RNase domains of the back-to-back dimers. Phosphorylation of the kinase domains of a dimer's constituent protomers may occur either sequentially, via dissociation and reassociation of tetramers, or simultaneously, via the assembly of multiple dimers into a larger oligomer. The unimpeded ability of K599A to form dimers and oligomers further supports the notion that phosphorylation takes place at the level of higher order oligomers rather than dimers.

Our results raise the question of whether the larger oligomers represent the maximally active form of IRE1 or are instead a phosphorylation-competent transient state en route to the generation of fully active phosphorylated dimers with and perhaps outside of oligomers. Our data showing the fast assembly and disassembly of high-order oligomers in response to saturating Tg, coupled with rapid hyper-phosphorylation, suggest that phosphorylated IRE1 may not need to remain oligomeric for its RNase to remain active. In fact, extensive phosphorylation either on the activation loop or elsewhere on IRE1's kinase domain may serve as a negative feedback mechanism, as previously proposed for yeast IRE1 (*Rubio et al., 2011*). A simple potential model for such negative feedback is that electrostatic repulsion across the face-to-face interface of two hyperphosphorylated dimers within an oligomer may break them apart into active back-to-back dimers and render the phosphate groups more accessible to the action of regulatory phosphatases (*Chang et al., 2018*).

Our data can be reconciled with seemingly contradictory earlier work to build a more comprehensive understanding of IRE1's biology. First, the single-particle tracking approach does not rule out the existence of monomeric IRE1; it is entirely plausible that a small fraction of monomers remain in equilibrium with a largely dimeric resting population. Transient monomerization of the IF1$^L$ interface, for example through the action of ER-lumenal chaperones (*Amin-Wetzel et al., 2017*), may play an important role in the regulated attenuation of IRE1 signaling in response to prolonged stress as previously suggested (*Pincus et al., 2010*). Conversely, we do not claim that mammalian IRE1 lacks the capacity to assemble into clusters larger than tetramers; as with any oligomerization-prone protein, such clustering may largely be a function of protein expression level. The substantially lower expression levels achieved in this study are likely the reason that we did not observe endogenously tagged IRE1 assembling into large clusters. It is clear that the lack of large cluster formation is not a defect of the HaloTag construct, since the same construct readily assembles into stress-induced puncta when overexpressed (*Figure 1—figure supplement 3*). Conversely, we can rule out the possibility that IRE1 clustering is an artifact of fluorescent protein fusions. In a parallel study by Gómez-Puerta et al., the authors expressed IRE1-GFP in HeLa cells at near-endogenous levels and demonstrated that it remains fully active despite not forming microscopically detectable clusters (*Gómez-Puerta et al., 2021*). Furthermore, in an earlier study from our own lab, we found that nearly one half of cells overexpressing an IRE1-mNeonGreen reporter did not form clusters over the time course of stress, despite fully splicing *XBP1* mRNA (*Belyy et al., 2020*). We also do not contend that large IRE1 clusters artificially result from overexpression: IRE1 levels were found to be highly variable across a panel of cancer cell lines (*Harnoss et al., 2019*; *Harnoss et al., 2020*) and it is reasonable to suspect that large clusters of endogenous IRE1 do form in cell lines with elevated amounts of IRE1 protein.

Rather than entirely ruling out the existence of IRE1 monomers or large oligomers, the present study demonstrates that IRE1 has the propensity to preassemble into inactive dimers in the absence of stress induction and that oligomerization past the tetrameric state is not strictly required for its RNase activation. In solution reactions, kinase/RNase dimers are capable of performing stem-loop endomotif-specific mRNA cleavage, while phosphorylated oligomers perform this function more efficiently than dimers and acquire a more promiscuous RNase activity termed RIDDLE (*Le Thomas et al., 2021*). The present data suggest that in a cellular context, non-phosphorylated full-length IRE1 dimers are more restricted, perhaps by orientation, while even endomotif-directed RNase activity requires IRE1 oligomerization and phosphorylation. This restriction may be simply steric, due to the limited ability of the unphosphorylated membrane-tethered kinase/RNase dimers to adopt a conformation conducive to RNA cleavage, or it may arise from IRE1's association with supplementary molecular players such as the ribosome (*Acosta-Alvear et al., 2018*) and/or the Sec61 translocon (*Sundaram et al., 2017*). Additionally, it remains to be determined how other signals—such as lipid saturation, which has been found to activate IRE1 independently of its lumenal domain (*Volmer et al., 2013*)—impact IRE1's oligomerization and phosphorylation states. Based on prior work in the field, it is likely that changes in

the ER membrane's lipid composition directly drive IRE1 molecules into phosphorylation-competent small oligomers via key features of IRE1's transmembrane helix (*Kono et al., 2017*).

Protein oligomerization is a universal mechanism that enables biological communication both between and within cells. Yet, experimental approaches for interrogating subtle oligomeric changes in the intracellular milieu remain scarce and fraught with caveats. We have developed a highly sensitive approach to detect differences between monomers, dimers, and small oligomers in the plane of the ER membrane. Our strategy is straightforward to implement and resilient to researcher bias, owing to its automated data analysis. Applying this approach to a key regulator of cellular proteostasis, IRE1, we demonstrated that the dimer-to-oligomer transition serves as the primary regulatory step in enzymatic activation, and reinforced the role of the lumenal domain as the master governor of IRE1's oligomeric state. IRE1 has emerged as a highly promising molecular target in an ever-growing list of human diseases. Uncovering the basic principles underlying its regulation promises to advance the design of future therapeutics, especially those intended to tune IRE1 activity through modulation of its oligomeric state.

# Materials and methods

## Key resources table

| Reagent type (species) or resource | Designation | Source or reference | Identifiers | Additional information |
|---|---|---|---|---|
| Gene (*Homo sapiens*) | ERN1 | HUGO Gene Nomenclature Committee | HGNC:3,449 | Gene encoding human IRE1a |
| Strain, strain background (*Escherichia coli*) | Stellar Competent Cells | Takara Bio | 636,763 | High-efficiency competent cells for cloning |
| Cell line (*Homo-sapiens*) | U-2 OS WT Flp-In T-REx | Ivan Dikic lab; verified by ATCC STR service; published in *Belyy et al., 2020* | PWM253 | Parental line for all cells in this study, denoted as "WT" |
| Cell line (*Homo-sapiens*) | U-2 OS IRE1α KO | *Belyy et al., 2020* | PWM254 | CRISPR knock-out of IRE1α in PWM253 |
| Cell line (*Homo-sapiens*) | U-2 OS IRE1α partial KO | This paper | PWM359 | Partial CRISPR knock-out of IRE1α in PWM253, containing one intact ERN1 allele |
| Cell line (*Homo-sapiens*) | U-2 OS IRE1a-HaloTag endogenously tagged | This paper | PWM360 | Introduction of a C-terminal HaloTag into the endogenous ERN1 locus of PWM359 cells. Clonal population. |
| Cell line (*Homo-sapiens*) | U-2 OS IRE1a-HaloTag endogenously tagged, low expression clone | This paper | PWM361 | Introduction of a C-terminal HaloTag into the endogenous ERN1 locus of PWM359 cells. clonal population with lower IRE1 expression level than PWM360. |
| Recombinant DNA reagent (plasmid) | SpCas9 and gRNA targeting the C-terminus of HsIRE11 | This paper | pPW3754 | Used for endogenous CRISPR editing of ERN1 gene. |
| Recombinant DNA reagent (plasmid) | HDR-HsIRE1a-10xGS-HaloTag | This paper | pPW3755 | Used for endogenous CRISPR editing of ERN1 gene. |
| Recombinant DNA reagent (plasmid) | CMVd3-ERmembrane-HaloTag-KKMP | This paper | pPW3756 | Used for transient transfections |
| Recombinant DNA reagent (plasmid) | CMVd3-ERmembrane-2xHaloTag-KKMP | This paper | pPW3757 | Used for transient transfections |
| Recombinant DNA reagent (plasmid) | CMVd3-HsIRE1-HaloTag | This paper | pPW3758 | Used for transient transfections |
| Recombinant DNA reagent (plasmid) | CMVd3-HsIRE1deltaLD-HaloTag | This paper | pPW3759 | Used for transient transfections |
| Recombinant DNA reagent (plasmid) | CMVd3-HsIRE1-K599A_KinaseDead-HaloTag | This paper | pPW3760 | Used for transient transfections |
| Recombinant DNA reagent (plasmid) | CMVd3-HsIRE1(WLLI-GSGS) [359-362]-HaloTag | This paper | pPW3761 | Used for transient transfections |
| Recombinant DNA reagent (plasmid) | CMVd3-HsIRE1(K121Y)-HaloTag | This paper | pPW3762 | Used for transient transfections |

*Continued on next page*

*Continued*

| Reagent type (species) or resource | Designation | Source or reference | Identifiers | Additional information |
|---|---|---|---|---|
| Recombinant DNA reagent (plasmid) | CMVd3-HsIRE1dLKR-HaloTag | This paper | pPW3763 | Used for transient transfections |
| Recombinant DNA reagent (plasmid) | CMVd3-ERmembrane-GST-HaloTag-KKMP | This paper | pPW3781 | Used for transient transfections |
| Recombinant DNA reagent (plasmid) | CMVd3-ERmembrane-GST-2xHaloTag-KKMP | This paper | pPW3783 | Used for transient transfections |
| Antibody | Anti- IRE1α (Rabbit monoclonal) | Cell Signaling Technology | 3294 | WB (1:1000) |
| Antibody | Anti- PERK (Rabbit monoclonal) | Cell Signaling Technology | 3192 | WB (1:1000) |
| Antibody | Anti- ATF4 (Rabbit monoclonal) | Cell Signaling Technology | 11,815 | WB (1:1000) |
| Antibody | Anti- CHOP (Mouse monoclonal) | Cell Signaling Technology | 2895 | WB (1:1000) |
| Antibody | Anti- ATF6 (Mouse monoclonal) | Proteintech | 66563–1 | WB (1:1000) |
| Antibody | Anti- β-actin (Rabbit monoclonal) | Cell Signaling Technology | 5125 | WB (1:1000) |
| Antibody | Anti-IRE1p (Rabbit monoclonal) | *Chang et al., 2018* | N/A | WB (1:1000) |
| Antibody | Anti-XBP1s (Rabbit monoclonal) | *Chang et al., 2018* | N/A | WB (1:1000) |
| Sequence-based reagent | Hs00176385_m1 | ThermoFisher Scientific | | qPCR primer for IRE1 |
| Sequence-based reagent | Hs02856596_m1 | ThermoFisher Scientific | | qPCR primer for XBP1u |
| Sequence-based reagent | Hs03929085_g1 | ThermoFisher Scientific | | qPCR primer for XBP1s |
| Sequence-based reagent | Hs01045913_m1 | ThermoFisher Scientific | | qPCR primer for DGAT2 |
| Sequence-based reagent | Hs00170663_m1 | ThermoFisher Scientific | | qPCR primer for BCAM |
| Sequence-based reagent | Hs00197728_m1 | ThermoFisher Scientific | | qPCR primer for TGOLN2 |
| Sequence-based reagent | s02800695_m1 | ThermoFisher Scientific | | qPCR primer for HPRT1 |
| Commercial assay or kit | TaqMan RNA-to-CT 1-Step Kit | ThermoFisher Scientific | 4392938 | |
| Commercial assay or kit | RNeasy Plus kit | Qiagen | 74,134 | |
| Commercial assay or kit | In-Fusion HD Cloning | Clontech | Clontech:639,647 | |
| Software, algorithm | saSPT | *Heckert et al., 2021* | https://github.com/alecheckert/saspt | Open-source software package used for extracting diffusion coefficients from single-particle trajectories. |
| Other | JF549 dye conjugated with HaloTag ligand | Luke Lavis Lab; *Grimm et al., 2017* | Promega: GA1110 | Kind gift of Luke Lavis; also available commercially from Promega |
| Other | JF646 dye conjugated with HaloTag ligand | Luke Lavis Lab; *Grimm et al., 2017* | Promega: GA1120 | Kind gift of Luke Lavis; also available commercially from Promega |

## Cell culture and experimental reagents

U-2 OS Flp-In T-REx cells were a kind gift of the Ivan Dikic lab and were independently authenticated through the human STR profiling service offered by the American Type Culture Collection (ATCC). Cells were cultured at 37 °C with 5% $CO_2$ in high-glucose DMEM (Thermo Fisher) supplemented with 10% tetracycline-free fetal bovine serum (FBS; Takara Bio), 6 mM L-glutaminfe, and 100 U/ml penicillin/streptomycin. All cell lines used in the study tested negative for mycoplasma contamination when assayed with either the Universal Mycoplasma Detection Kit (ATCC 30–1012 K) or the MycoAlert Detection Kit (Lonza LT07-418). Tunicamycin and thapsigargin were purchased from Sigma-Aldrich or from Tocris and dissolved in DMSO. JF549 and JF646 dyes conjugated with the HaloTag ligand were a kind gift of Luke Lavis (Janelia Farms). The antibodies used for immunoblotting are listed in the *Immunoblotting* section.

## Endogenous tagging of IRE1 in U-2 OS cells

To achieve full editing despite the hyperploid nature of the U-2 OS cells, we first generated a partial IRE1α knockout cell line harboring a single intact allele of *ERN1*, the gene encoding IRE1α (cell line ID: PWM359). This was done using the same CRISRP/Cas9-based approach that we used to generate

a complete IRE1α knockout in our previous paper (*Belyy et al., 2020*), except that rather than looking for a clone that contained no copies of WT *ERN1*, we identified clones that contained a single unedited allele. The presence of a single intact *ERN1* allele was confirmed by TOPO cloning and immunoblotting. These partial knock-out cells were then co-transfected with a plasmid encoding Cas9 with the guide RNA and a homology-directed repair (HDR) template plasmid targeted at C-terminus of *ERN1*. Design of both plasmids followed the protocol published elsewhere (*McKinley, 2018*). Edited cells were selected by fluorescence-activated cell sorting (FACS), separated into clonal populations by limiting dilution, and assayed for IRE1α expression and UPR activation by immunoblotting and RT-PCR. Two clones were selected for further study: a somewhat higher expressing clone (cell line ID: PWM360) and a somewhat lower expressing clone (cell line ID: PWM361). When immunoblotted against IRE1α, both clones produced a clear band that ran slower than WT IRE1α, indicating a successful integration of the full-length HaloTag.

## Sample preparation for microscopy

Cells were seeded at a density of $1.6 \times 10^4$ cells/cm$^2$ into glass-bottom 8 well chamber slides (ibidi 80827), which were pre-coated with rat tail collagen type I (Corning 354236) at 10 μg/cm$^2$ in accordance with the manufacturer's instructions (briefly, a 2 hr incubation at room temperature). Twenty-four hours prior to imaging, the growth medium was replaced with 'Imaging medium': FluoroBrite DMEM (ThermoFisher) supplemented with 10% tetracycline-free fetal bovine serum (FBS; Takara Bio) and 6 mM L-glutamine, without antibiotics. For experiments requiring transfection, cells were transfected with a mixture of 50 ng of plasmid DNA and 50 ng of carrier salmon sperm DNA per well immediately following medium change (i.e. 24 hr prior to the start of imaging). Transfections were carried out in "Imaging medium" using the Fugene HD transfection reagent (Promega).

On the day of imaging, cells were treated with ER stressors at the indicated time points. Labeling with JF549 and JF646 dyes conjugated with the HaloTag ligand was initiated 1.5 hr prior to the start of imaging. First, the dyes were added to pre-warmed 'Imaging medium', and this medium was used to replace the cells' growth medium. We experimentally found the optimal molar dye ratio to achieve ~50% labeling with each ligand to be 1:20 (5 nM JF549-HaloTag and 100 nM JF646-HaloTag). The large difference in required concentrations is likely due to the difference in membrane permeability between the two dyes. We experimentally found the 5 nM JF549 /100 nM JF646 concentrations to be saturating under our labeling conditions since further increases in dye amounts did not lead to a further increase in the density of diffusing spots in IRE1-HaloTag cells. Following addition of the medium containing the two dyes (and any required ER stressors), cells were returned to the incubator for 1 hr. Then, cells were washed twice with warm PBS, washed once with pre-warmed 'Imaging medium', and returned to the incubator for an additional 5 min to give any unbound dye time to diffuse out of the cells. The medium was replaced one more time with pre-warmed 'Imaging medium' containing any required ER stressors to finish sample preparation.

## Microscopy

All imaging was carried out on one of two Nikon Ti-E inverted microscopes (#1 and #2 hereafter), each equipped with a Nikon motorized TIRF module, an Agilent/Keysight MLC400 fiber-coupled laser light source, a Perfect Focus System (PFS, Nikon), a 100 × 1.49 NA oil immersion objective (Apo TIRF, Nikon), and a Hamamatsu Flash 4.0 CMOS camera. Microscope #1 held a ZET405/488/561/640m-TRFv2 quadruple bandpass filter cube (Chroma), while microscope #2 held a ZET488/561/640 m triple bandpass filter cube (Chroma). Additionally, microscope #1 included a Yokogawa CSU-X high-speed confocal scanner unit and an Andor iXon 512 × 512 EMCCD camera, which were used for spinning-disk confocal microscopy experiments. Both microscopes featured full temperature and $CO_2$ control to maintain the samples at 37 °C and 5% $CO_2$, one using a custom-built enclosure (#1) and the other using an OkoLab Live stage insert (#2). All components of microscope #1 were controlled by the μManager open-source platform (*Edelstein et al., 2010*), while microscope #2 was controlled with NIS-Elements software (Nikon).

Oblique-angle illumination conditions were achieved by focusing on a cell, engaging the PFS, and gradually increasing illumination angle with the motorized TIRF lens until single-molecule spots along the bottom surface of the cell became clearly visible. Videos were acquired with a 60ms combined frame time, split into a 25ms exposure in the JF549 channel (561 nm laser, operated at 25 mW) and

a 25ms exposure in the JF646 channel (640 nm laser, operated at 40 mW), with the remaining 10ms accounting for channel switching times. The two channels were imaged sequentially by the same camera using camera-triggered switching of the acousto-optic tunable filter (AOTF) built into the light source. Frames were cropped to approximately 500 × 500 pixels prior to acquisition since the full camera sensor could not be read out fast enough to support the required frame rate. Typically, 100 combined frames were acquired per cell (6 s total movie duration), which in our hands provided a good number of trajectories per cell while avoiding extensive photobleaching of the dyes. To locate and choose cells for imaging, we used the full size of the camera sensor and acquired a series of tiled snapshots of an area containing ~100 cells. We then selected cells that were morphologically normal, well-adhered, and spread out. When imaging transiently transfected cells, we chose cells in which the HaloTag-labeled proteins were expressed at sufficiently low levels to allow us to clearly see individual spots corresponding to single molecules.

## Data analysis

Single-molecule data were analyzed to identify co-localizing two-color trajectories using a pipeline developed in house, described in detail below. First, each movie was split into the two individual channels, JF549 and JF646. Next, spots were located using the Laplacian of Gaussian (LoG) detector implemented in the TrackMate plugin (*Tinevez et al., 2017*) for ImageJ. Then, identified spots were tracked using the Linear Assignment Problem (LAP) algorithm (*Jaqaman et al., 2008*), also implemented within the TrackMate plugin. All input parameters for both the LoG detector and the LAP tracker were chosen empirically to match the expected output in a subset of randomly selected single-molecule movies; afterwards, they were kept constant for the analysis of all data used to construct the plots presented in this paper. To speed up analysis and ensure that the exact same settings are used to process every movie, we scripted TrackMate to read all settings from a standardized JSON configuration file and perform both spot detection and tracking on all movie files contained within a given folder. The TrackMate output files containing spot and track data for each channel were then saved to disk in the XML format for further analysis.

All subsequent analysis was performed in Python. The broad goal of this analysis was to identify tracks that correlated well in space and time between the JF549 and JF646 channels. To avoid problems imposed by uneven track durations and trajectories crossing each other, we decided to perform the analysis using a short sliding window. In other words, instead of considering the entire movie at once, we binned each movie into overlapping shorter movies containing a fixed number of frames each, and looked for co-localizing trajectories in each of the shorter movies. To achieve this, the TrackMate output files were parsed and filtered to only include tracks that span at least as many frames as the length of the sliding window. The sliding window was then moved across the duration of the movie in 1 frame increments. In each of the resulting windows, only tracks that were fully defined within that window (i.e. had position information for each frame) were selected for correlation analysis.

Pearson's correlation coefficients (PCC) were then individually calculated for the X- and Y-coordinates of every pair of spatially adjacent tracks (adjacent meaning that at least a subset of data points of track B are contained within the rectangle that bounds track A). The requirement for tracks being spatially adjacent both increased the computational efficiency of the algorithm and helped eliminate false positives from short tracks of similar shapes that occurred by chance in different parts of the cell. A pair of tracks was determined to be correlated if all of the following conditions were met: 1) the two tracks share at least one window $N$ frames long in which the position of each spot is well-defined in every frame, 2) At least one such window yields a PCC value greater than $T$ for both the X- and Y-coordinates of the tracked spot, and 3) The two tracks are at least partially overlapping in space. The value plotted in the figures, '% correlated trajectories', is defined as follows:

$$\% \ correlated \ trajectories \equiv \frac{n_{JF549corr}}{n_{JF549total}} * 100 \ ,$$

where $n_{JF549total}$ is the total number of trajectories in the JF549 channel that are at least as long as the length of the sliding window ($N$), while $n_{JF549corr}$ is the number of trajectories in the JF549 channel that are found to have correlated trajectories in the JF646 channel as described above.

In our analysis, the only user-selected parameters that tune the sensitivity of the approach are $N$ (the length of the window, in frames) and $T$ (the threshold value for the PCC). Just as with the tracking algorithm, we first empirically found values of $N$ and $T$ that yielded robust identification of visually

correlated tracks without giving too many false positives, and then used these values in all subsequent data analysis. We did find that due to differences in filters, laser intensity, and alignment between the two microscopes, a different combination of $N$ and $T$ yielded the highest dynamic range in our assay. Data from each microscope were fully internally consistent but we avoided showing data collected on two different microscopes on the same plot. Thus, each panel in the paper contains either data collected exclusively on microscope #1 or on microscope #2.

To speed up data processing and enhance reproducibility, we again scripted the analysis to read a single JSON configuration file that specifies the $N$ and $T$ parameters, along with a full list of folders containing the TrackMate XML files for each condition. The code reports the fraction of correlated tracks for each condition with 95% confidence intervals determined by bootstrapping. The README. md file included with the source code (*Belyy, 2022*) contains detailed instructions for running this analysis and replicating all plots in the paper from source data (*Belyy, 2021a*; *Belyy, 2021b*). In organizing the analysis software, we sought to make reproducing our data and adapting the code to different single-molecule co-localization studies as straightforward as possible.

## Measurement of diffusion coefficients from single-molecule data

Apparent diffusion coefficients were obtained from single-particle trajectories from tracking data using one of two methods: mean squared displacement (MSD) or state array (SA). In MSD analysis, each single-particle trajectory was broken up into 5-frame segments that were then aligned and averaged to obtain the final MSD plot for that particle. Particles that were tracked for fewer than 15 frames were excluded from MSD analysis. Each MSD plot was then fitted to a straight line($<r^2 \geq 4$ Dt) to determine the respective particle's diffusion coefficient. SA analysis was performed as described in the original manuscript (*Heckert et al., 2021*) through a direct implementation of the 'saspt' Python module. Additional analysis details for both approaches can be found in the README.md file included with the source code(*Belyy, 2022*).

## Estimation of IRE1 cluster stoichiometry

A simple yet useful model for estimating cluster stoichiometry based on the fraction of correlated tracks works as follows. Assume that each HaloTag-conjugated protein can occupy one of three states: bound to an unbleached JF549 dye molecule, bound to an unbleached JF646 dye molecule, or undetectable. The latter category is a catch-all for every possible reason a protein may escape detection such as dye bleaching, incomplete labeling, new protein synthesis after labeling reaction, and false negatives in the spot detection algorithm. Let the probabilities of these three states be denoted as $P_1$ (JF549-bound), $P_2$ (JF646-bound), and $P_u$ (undetectable). Because the combined probabilities must add up to unity, $P_u = 1 - P_1 - P_2$. Then, for a cluster comprised of $n$ individual molecules, we can express the total probability that the cluster contains at least one dye of each color as follows:

$$P_{both} = 1 - P_{JF549\ only} - P_{JF646\ only} - P_{no\ dye} =$$
$$= 1 - (1 - P_1)^n - (1 - P_2)^n + (1 - P_1 - P_2)^n$$

However, in our experiment, the clusters containing no detectable dyes are invisible, and what we measure experimentally is instead the observed fraction of all visible clusters that contain at least one dye of each color. Let's call this quantity the fraction of observed co-localizers, $F_{obs}$:

$$F_{obs} = \frac{P_{both}}{1 - P_{no\ dye}} =$$
$$= \frac{1 - (1 - P_1)^n - (1 - P_2)^n + (1 - P_1 - P_2)^n}{1 - (1 - P_1 - P_2)^n}$$

To estimate cluster stoichiometry based on the experimentally measurable $F_{obs}$, we first need a measurement of $P_1$ and $P_2$, which can be done using data from the constitutive 2 x HaloTag homodimer construct, where we know that n = 2. Let's assume that $P_1 = P_2 = P_L$ (labeling probability), since all our experiments are done in a regime where the labeling densities with the two different dyes are nearly identical. Plugging these assumptions into the expression for $F_{obs}$, we obtain:

$$F_{obs} = \frac{1 - 2(1 - P_L)^2 + (1 - 2P_L)^2}{1 - (1 - 2P_L)^2} = \frac{P_L}{2(1 - P_L)}$$

Rearranging this expression, we find that $P_L$ can be expressed in terms of $F_{obs}$:

$$P_L = \frac{2F_{obs}}{2F_{obs}+1}$$

Once we have an estimate of $P_L$ from the 2 x homodimer control (in our experiment, this value is typically around 0.14), it can be simply plugged into the earlier expression for $F_{obs}$ and plotted as a function of $n$ to yield an estimate of average cluster stoichiometry for any value of $F_{obs}$. Of course, this model is a significant oversimplification of the true underlying processes (mainly due to lumping all possible sources of error into the single term $P_u$), but it does provide a useful ballpark estimate.

## Detection of large IRE1 clusters

While we did not observe the formation of stress-induced IRE1 clusters in cells expressing IRE1-HaloTag from the endogenous genomic locus, some small punctate structures were occasionally visible in confocal images of these cells both with and without stress. Such small puncta were also present in some cells expressing HaloTag control constructs. We attribute these puncta to autofluorescence, local enrichment of ER membrane components in organelles such as lysosomes or autophagosomes, and/or dense regions of dense ER folds that cannot be adequately resolved by confocal microscopy. We sought to distinguish between these spots and the visually distinct "classic" IRE1 clusters (as described by us and others) by developing a quantitative definition of IRE1 clustering. We define a cell as exhibiting IRE1 clustering if greater than 1% of the integrated IRE1 fluorescence intensity is contained within small punctate structures. To identify such structures automatically, the image is first smoothened with a Gaussian filter with a sigma of 1 pixel. Then, the image is background-corrected and its intensity normalized such that the mean pixel intensity becomes 1. The normalized image is then processed with a Laplacian filter with a 3x3 pixel kernel to identify regions of maximal contrast and thresholded with an empirically found value of 5. Any areas contained by pixels that pass this threshold cutoff are considered to be potential puncta. These areas are then dilated with a 3x3 pixel mask and the combined fluorescence intensity contained in the potential puncta is compared against the total fluorescence intensity of the cell. If the tentative puncta contain more than 1% of the cell's total fluorescence, we conclude that IRE1 clusters are present in the cell. We find that these criteria robustly pass the visual test for a range of images across different illumination intensities and magnification levels.

## XBP1 mRNA splicing assays

Cells were grown in wells of a 12-well plate, treated with ER stressors as indicated in the figure, and harvested at ~70% confluency with TRIzol (Thermo Fisher) in accordance with the manufacturer's instructions. RNA was then extracted from the aqueous phase using a spin column-based purification kit (RNA Clean & Concentrator-5, Zymo Research # R1015) and reverse-transcribed into cDNA using SuperScript VILO Master Mix (Thermo Fisher # 11755050). The cDNA was diluted 1:10 and used as a template for PCR with the following primer pair: VB_pr259 (CGGAAGCCAAGGGGAATGAA) and VB_pr167 (ACTGGGTCCAAGTTGTCCAG). PCR was carried out with Taq polymerase (Thermo Fisher # 10342020) in the manufacturer-supplied Taq buffer supplemented with 1.5 μM Mg$^{2+}$. The following PCR program was used: (*Gurevich and Gurevich, 2018*) Initial denaturation: 95 °C for 2 min (*Shattil and Newman, 2004*), 95 °C for 30 s (*Chung, 2017*), 60 °C for 30 s (*Reich et al., 1997*), 72 °C at 30 s (*Ashkenazi and Dixit, 1998*). Repeat steps 2–4 27 more times, for 28 total PCR cycles. PCR products were visualized on a 3% agarose gel stained with SYBR Safe (Thermo Fisher S33102) and imaged on a ChemiDoc gel imaging system (BioRad).

## Immunoblotting and RT-qPCR

Cells were grown in 6-well plates in RPMI1640 or DMEM media supplemented with 10% (v/v) FBS (Sigma), 2 mM glutamine (Gibco), and 100 U/mL penicillin plus 100 μg/mL streptomycin (Gibco), and treated as indicated. Thapsigargin (Tocris) was used at a concentration of 100 nM and tunicamycin (Tocris) at 5 μg/mL, dissolved in DMSO. DMSO was used as the untreated control. When 70–80% confluent, cells were washed in PBS and trypsinized using Trypsin-EDTA 0.05%. Cells were pelleted and stored at –20 for protein or RNA extraction.

For immunoblotting, protein lysates were extracted in RIPA buffer (EMD Millipore) with Halt protease and phosphatase inhibitor cocktail (Thermo Scientific). The crude lysates were cleared by

centrifugation at 13,000 rpm for 15 min and protein content was analyzed by Pierce BCA protein assay (Thermo Scientific).

Equal amounts of protein (40 µg/condition) were run with SDS-PAGE and electrotransferred onto membranes that were blocked with 5% dried nonfat milk powder in TBST (blocking solution). Blots were incubated with 1/1000 dilution in blocking solution of primary antibodies overnight at 4°C. Antibodies (Abs) for IRE1α (3294), PERK (3192), ATF4 (11815), CHOP (2895) were from Cell Signaling Technology (CST). ATF6 antibody (66563–1) was from Proteintech. β-actin (5125) from CST was used as a housekeeping control. Abs for XBP1s and pIRE1 were generated at Genentech and have been described elsewhere (*Chang et al., 2018*). Blots were washed in TBST, then incubated during 1 h at room temperature with 1/10,000 dilution of the corresponding peroxidase-conjugated secondary antibodies in blocking solution: donkey anti-rabbit and anti-mouse from Jackson Immunoresearch. Blots were finally washed in TBST and analyzed using Super Signal West Dura or Femto (Thermo Scientific).

For RT-qPCR, RNA extraction was performed with the RNeasy Plus kit (Qiagen #74134). 50 ng of RNA per sample, in technical triplicates, were reverse transcribed and amplified on the ABI Quant-Studio 7 Flex Real-Time PCR System, using TaqMan RNA-to-CT 1-Step Kit (ThermoFisher Scientific #4392938). The RQ (relative quantification, 2-ΔΔCt) was calculated by relating each individual CT value to the expression of the housekeeping HPRT1 gene and the control of the experiment. Taqman primers for IRE1 (Hs00176385_m1), XBP1u (Hs02856596_m1), XBP1s (Hs03929085_g1), DGAT2 (Hs01045913_m1), BCAM (Hs00170663_m1), TGOLN2 (Hs00197728_m1), and HPRT1 (Hs02800695_m1) were from ThermoFisher Scientific.

## Statistics

All single-particle tracking data are reported as mean values of cell-by cell percent correlated trajectories (as defined in the *Data Analysis* section) with 95% confidence intervals. For pairwise comparisons of individual conditions, p-values for each pair of conditions were calculated both by two-tailed t-test and by an approximate permutation test with 10,000 iterations. Tables containing all pairwise p-values are included with every figure as source data. Individual data points are overlaid on the graphs, with each point representing the percent of correlated trajectories in one cell. Data for individual conditions were pooled from multiple experiments conducted on separate days (biological replicates). The exact data points and raw files going into each condition can be obtained from the human-readable JSON files that are published alongside the manuscript. The only files that were excluded from analysis were movies in which the cell was clearly out of focus, TIRF illumination was clearly out of alignment, or the expression level of the transfected construct was so high that individual spots were not resolvable. Otherwise, no data were excluded and no outliers were removed. No explicit power analysis to determine sample size was conducted prior to the start of the experiment; instead, sample sizes were chosen to comprise no fewer than ten individual cells (in most cases, many more), corresponding to ~100,000 or more individual particle trajectories.

## Availability of materials, data, and software

The code used to analyze raw data and generate all figures in this paper is freely available from Zenodo (*Belyy, 2022*). All raw single-molecule microscopy data are available from Dryad (*Belyy, 2021a*). All other raw data, including full gel images, together with processed single-molecule microscopy data, are available from Zenodo (*Belyy, 2021b*). All cell lines and constructs used in this paper are available upon request.

## Acknowledgements

We thank members of the PW and A Ashkenazi laboratories for helpful discussions, especially Smriti Sangwan, Tsan-wen Lu, Adrien Le Thomas, David A Lawrence, Silvia Ramundo, and Morgane Boone. We thank Nico Stuurman for advice and help with single-molecule microscopy and Elif Karagöz (Max Perutz Labs, Universität Wien, Vienna, Austria) for comments on the manuscript. We thank David Ron (Cambridge Institute for Medical Research, University of Cambridge, Cambridge, United Kingdom) for proposing the HaloTag-GST dimerization control. We thank Ivan Dikic (Institute of Biochemistry II, School of Medicine, Goethe University, Frankfurt am Main, Germany) and Luke Lavis (Janelia Research Campus, Howard Hughes Medical Institute, Ashburn, VA, USA), for reagents and advice.Funding This

research was supported by NIH K99-GM138896 (to VB) and NIH R01-GM032384 (to PW). PW is an Investigator of the Howard Hughes Medical Institute. VB is a Damon Runyon Fellow supported by the Damon Runyon Cancer Research Foundation (DRG-2284–17).

## Additional information

### Competing interests

Iratxe Zuazo-Gaztelu, Avi Ashkenazi: This author was an employee of Genentech, Inc during performance of this work. The other authors declare that no competing interests exist.

### Funding

| Funder | Grant reference number | Author |
|---|---|---|
| National Institute of General Medical Sciences | R01-GM032384 | Peter Walter |
| National Institute of General Medical Sciences | K99-GM138896 | Vladislav Belyy |
| Howard Hughes Medical Institute | | Peter Walter |
| Damon Runyon Cancer Research Foundation | DRG-2284-17 | Vladislav Belyy |

The funders had no role in study design, data collection and interpretation, or the decision to submit the work for publication.

### Author contributions

Vladislav Belyy, Conceptualization, Data curation, Formal analysis, Funding acquisition, Investigation, Methodology, Software, Validation, Visualization, Writing – original draft, Writing – review and editing; Iratxe Zuazo-Gaztelu, Formal analysis, Investigation, Methodology, Validation, Writing – review and editing; Andrew Alamban, Formal analysis, Investigation, Software, Writing – review and editing; Avi Ashkenazi, Conceptualization, Project administration, Resources, Supervision, Writing – review and editing; Peter Walter, Conceptualization, Funding acquisition, Project administration, Resources, Supervision, Writing – review and editing

### Author ORCIDs

Vladislav Belyy ![ORCID] http://orcid.org/0000-0003-2813-8215
Avi Ashkenazi ![ORCID] http://orcid.org/0000-0002-6890-4589
Peter Walter ![ORCID] http://orcid.org/0000-0002-6849-708X

### Decision letter and Author response

Decision letter https://doi.org/10.7554/eLife.74342.sa1
Author response https://doi.org/10.7554/eLife.74342.sa2

## Additional files

### Supplementary files

- Supplementary file 1. Table containing detailed information for all plasmids used in this study.
- Supplementary file 2. Full plasmid maps for all plasmids used in this study.
- Transparent reporting form

### Data availability

Raw single-particle data were deposited to Dryad: https://doi.org/10.5061/dryad.t4b8gtj33. Processed and additional raw data were deposited to Zenodo with the following DOI: https://doi.org/10.5281/zenodo.5513025. Analysis code was deposited to Zenodo with the following DOI: https://doi.org/10.5281/zenodo.5540055.

The following datasets were generated:

| Author(s) | Year | Dataset title | Dataset URL | Database and Identifier |
|---|---|---|---|---|
| Belyy V, Zuazo-Gaztelu I, Alamban A, Ashkenazi A, Walter P | 2021 | Raw microscopy data from: Endoplasmic reticulum stress activates human IRE1α through reversible assembly of inactive dimers into small oligomers | https:/doi.org/10.5061/dryad.t4b8gtj33 | Dryad Digital Repository, 10.5061/dryad.t4b8gtj33 |
| Belyy V | 2021 | Processed and additional data for our publication titled "Endoplasmic reticulum stress activates human IRE1α through reversible assembly of inactive dimers into small oligomers" | https://doi.org/10.5281/zenodo.5513025 | Zenodo, 10.5281/zenodo.5513025 |

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
