## [Editor Report]

In this study, Belyy et al., have developed a powerful new imaging system--one that will benefit others in the cell biology community--to measure how a key transducer of the unfolded protein response , Ire1, responds to endoplasmic reticulum (ER) stress. While prior studies indicated the existence of large Ire1 oligomers that arose in the ER membrane after stress, this study used single molecule tracking and native levels of Ire1 to demonstrate that Ire1 naturally exists in the inactive state as a dimer and that much smaller oligomers form after stress, a phenomenon governed by the Ire1 lumenal domain. Moreover, Ire1 trans-phosphorylation, which is required for activation, begins after oligomer formation. Overall, these studies yield unprecedented insight into the mechanism underlying the unfolded protein response and have revised our understanding of Ire1 dynamics.

---

## [Decision Letter]

**Decision letter after peer review:**

Thank you for submitting your article "Endoplasmic reticulum stress activates human IRE1α through reversible assembly of inactive dimers into small oligomers" for consideration by *eLife*. Your article has been reviewed by 3 peer reviewers, and the evaluation has been overseen by a Reviewing Editor and Vivek Malhotra as the Senior Editor. The reviewers have opted to remain anonymous.

Overall comments and conclusions.

As you will note in the associated files, your manuscript was reviewed by three experts in the field, and their comments are now available to you. We are delighted to report that the reviewers universally agreed that the strengths of this study include the development of an exciting and clever approach to probe membrane protein oligomerization, the use of a technological advance that will be applicable to studies of other protein substrates, the fact that the experiments were expertly performed and controlled, and the potential that this study provides a new advance in our understanding of the early and later events during UPR induction.

However, after much discussion, the reviewers universally and consistently hovered around the question of how to reconcile the conclusions from this study with prior work from your lab and others using fluorescently tagged IRE1 fusions. Thus, there was a general feeling that the current story needs to conclusively account for why select conclusions from earlier publications suffered from an inherent limitation. Additional, related comments included: (1) Better verification that the IRE-HALO protein remains active during the course of the experiment, specifically in the presence of ligands/fluorophores, since some downstream events after IRE1 activation were not investigated; (2) It appears that the cell type and expression levels seem to be unchanged between this study and prior studies, but this raises the question of whether there is another confounding factor. Two other questions also arose from the reviewers/discussion, including how this model explains IRE1 activation/oligomerization induced by lipid saturation, and whether there is a way to verify what a bona fide tetramer might look like using this new system. In short, is there a control substrate that can be examined?

In sum, there is a strong feeling that additional work is needed to better resolve what will be noted as a discrepancy in the field, and thus the results-if published as is-might muddy the waters.

We sincerely hope that the revisions will not prove overly burdensome, especially as the conclusions are novel and the method will be welcomed by many in the cell biology field.

*Reviewer #1 (Recommendations for the authors):*

1. An important control that appears missing is demonstrating that the IRE1-HALO tag remains active and functional under the same conditions used in the imaging. In other words, the authors should perform experiments showing that IRE1-HALO autophosphorylation and RNase activation are not impacted by the presence of the fluorophores used to map trajectories in the microscopy experiments. The presence of large fluorescent ligands could impact IRE1 activity, so it should be clearly demonstrated that this is not the case, especially in light of the comparisons between oligomerization and activity discussed.

2. Similar to the above, it would be good to have quantitation of markers of IRE1 activity (e.g., autophosphorylation and XBP1 splicing) to confirm the model whereby oligomerization precedes autophosphorylation/activation. Specifically, as indicated above, under the same conditions. This does appear accurate based on the presented data, but demonstrating this more explicitly would help support the manuscript and this important point more strongly.

3. I have some questions about the relationship between the work described herein and previous work monitoring IRE1 oligomerization using IRE1-GFP, specifically with respect to the mutant constructs that were overexpressed in this work. I understand the limited sensitivity of confocal made it difficult to detect small oligomers with IRE1-GFP, but I'm curious as to why the larger oligomeric clusters were not observed with the HALO-tagged IRE1 when ectopically expressed. I assume this is coming from lower levels of expression with the CMVd3 promoter, but it would be good to show that similar experiments with IRE1-GFP do not show the larger clusters observed in previous studies. Especially with respect to the fact that the model described herein contradicts these previous studies.

4. Lastly, the authors show that the luminal domain is primarily responsible for ER stress-dependent oligomerization of IRE1. However, lipid saturation can activate IRE1 through a mechanism that appears independent of its luminal domain. It would be good to include some potential discussion of how lipid saturation can bypass this oligomerization step to promote IRE1 oligomerization and/or autophosphorylation, especially because the presented results suggest IRE1 would be monomeric in the absence of the luminal domain.

*Reviewer #2 (Recommendations for the authors):*

Determining which form(s) of IRE1a are active and inactive in cells is fundamental to understanding the Unfolded Protein Response. A single particle tracking imaging approach is highly appropriate and probably the best approach. Unfortunately, some fundamental issues significantly dampen my enthusiasm for this manuscript.

The main issue with the manuscript is the lack of apparent clustering of the IRE1a-Halo reporter. Given all of the other studies that observe clustering with different IRE1a reporters with different fusions in different cell types, including multiple studies from the Walter lab, the lack of clustering raises the reasonable question of: What is being studied here? The IRE1a clusters and clustering behavior described in the recent Belyy et al., study would likely produce very different % correlated trajectory results. If the results in the current study are unique to the reporter, it's unclear what the relevance is to the rest of the ER stress field. To provide confidence in this system, I think the authors need to determine 1. why there is a discrepancy and 2. which system(s) accurately report on IRE1a behavior. I appreciate that these tasks represent a significant amount of work and I am not making these suggestions lightly.

Regarding the nature of the discrepancy, it likely revolves around the Halo Tag, the JF dyes, and/or the insertion site. Either the new reporter sets a new standard for an inert tag or the new reporter will raise a cautionary note for the Halo Tag system. One useful approach would be the addition of a minimal epitope tag to IRE1, such as an HA tag, and analysis of the reporter's behavior with immunofluorescence of stressed and unstressed cells. It is probably the most physiologic system, short of a great anti-IRE1a IF antibody. With such a reporter, please assess whether IRE1a-epitope tag clusters form in stressed and not in unstressed cells. If affirmed, then the authors will need to dissect and optimize the Halo Tag system. Note that an HA tag has been used in yeast and stress induced-Ire1 clustering was readily observed by Kimata et al. If no clusters are observed with stressed epitope-tagged mammalian IRE1a, it would be helpful for the ER stress field for the authors to determine how fluorescent proteins, even robust monomers such as mNG, impact clustering and whether a significant re-evaluation of IRE1a clustering models will be necessary.

*Reviewer #3 (Recommendations for the authors):*

Given that, previously, IRE1 structural rearrangements have been tied to the XBP1-splicing-versus-RIDD functionalities of IRE1, does the IRE1a-HALO activate RIDD to the same extent and with the same kinetics as endogenous?

The tracking of multimerization through movement raises two related questions. One is whether the automated visualization is then biased for movement, such that, if higher ordered structures of IRE1 were relatively immobile, they would be missed. The flip side of this coin is that, presumably, embedded within the already-existing data is information on whether ER stress changes the mobility of IRE1, comparing correlated tracks to non-correlated tracks, and comparing correlated tracks in the presence and absence of stress.

---

## [Author Response]

Overall comments and conclusions.As you will note in the associated files, your manuscript was reviewed by three experts in the field, and their comments are now available to you. We are delighted to report that the reviewers universally agreed that the strengths of this study include the development of an exciting and clever approach to probe membrane protein oligomerization, the use of a technological advance that will be applicable to studies of other protein substrates, the fact that the experiments were expertly performed and controlled, and the potential that this study provides a new advance in our understanding of the early and later events during UPR induction.However, after much discussion, the reviewers universally and consistently hovered around the question of how to reconcile the conclusions from this study with prior work from your lab and others using fluorescently tagged IRE1 fusions. Thus, there was a general feeling that the current story needs to conclusively account for why select conclusions from earlier publications suffered from an inherent limitation. Additional, related comments included: (1) Better verification that the IRE-HALO protein remains active during the course of the experiment, specifically in the presence of ligands/fluorophores, since some downstream events after IRE1 activation were not investigated; (2) It appears that the cell type and expression levels seem to be unchanged between this study and prior studies, but this raises the question of whether there is another confounding factor. Two other questions also arose from the reviewers/discussion, including how this model explains IRE1 activation/oligomerization induced by lipid saturation, and whether there is a way to verify what a bona fide tetramer might look like using this new system. In short, is there a control substrate that can be examined?In sum, there is a strong feeling that additional work is needed to better resolve what will be noted as a discrepancy in the field, and thus the results-if published as is-might muddy the waters.We sincerely hope that the revisions will not prove overly burdensome, especially as the conclusions are novel and the method will be welcomed by many in the cell biology field.

We thank the editors and all three reviewers for their helpful comments and feedback. In the revised version of the manuscript, we have addressed all comments, performed several additional experiments, and directly addressed the question of reconciling the conclusions of this manuscript with prior work from our lab and others. We hope that the reviewers agree that the manuscript has been substantially improved by the revision process and warrants publication in *eLife* in its new form.

Reviewer #1 (Recommendations for the authors):1. An important control that appears missing is demonstrating that the IRE1-HALO tag remains active and functional under the same conditions used in the imaging. In other words, the authors should perform experiments showing that IRE1-HALO autophosphorylation and RNase activation are not impacted by the presence of the fluorophores used to map trajectories in the microscopy experiments. The presence of large fluorescent ligands could impact IRE1 activity, so it should be clearly demonstrated that this is not the case, especially in light of the comparisons between oligomerization and activity discussed.

This is a very fair point. We have repeated the key biochemical experiments under conditions identical to those used for imaging (including not only labeling with fluorophores, but also growing cells on collagen-coated glass coverslips in clear FluoroBrite medium). The results, shown in the new Figure 4 —figure supplement 1, indicate that IRE1-HaloTag remains fully active under these conditions.

2. Similar to the above, it would be good to have quantitation of markers of IRE1 activity (e.g., autophosphorylation and XBP1 splicing) to confirm the model whereby oligomerization precedes autophosphorylation/activation. Specifically, as indicated above, under the same conditions. This does appear accurate based on the presented data, but demonstrating this more explicitly would help support the manuscript and this important point more strongly.

We agree with this point. While quantification of Western blots is still inherently imprecise, we performed qPCR assays to quantitatively monitor XBP1 splicing and RIDD activity of IRE1. These data are shown in the new Figure 1 —figure supplement 2.

3. I have some questions about the relationship between the work described herein and previous work monitoring IRE1 oligomerization using IRE1-GFP, specifically with respect to the mutant constructs that were overexpressed in this work. I understand the limited sensitivity of confocal made it difficult to detect small oligomers with IRE1-GFP, but I'm curious as to why the larger oligomeric clusters were not observed with the HALO-tagged IRE1 when ectopically expressed. I assume this is coming from lower levels of expression with the CMVd3 promoter, but it would be good to show that similar experiments with IRE1-GFP do not show the larger clusters observed in previous studies. Especially with respect to the fact that the model described herein contradicts these previous studies.

We thank the reviewer for raising this important point. The reviewer is correct in assuming that the CMVd3 promoter generally results in low expression levels of IRE1-HaloTag upon transient transfection (which is exactly why we chose to use this promoter), but the IRE1-HaloTag construct is fully capable of forming clusters when overexpressed. In fact, even with the CMVd3 promoter, we see a small fraction of cells that express elevated levels of IRE1-HaloTag; these cells readily form IRE1 puncta upon induction of ER stress. We are showing example confocal images of these cells in the new Figure 1 —figure supplement 3.

4. Lastly, the authors show that the luminal domain is primarily responsible for ER stress-dependent oligomerization of IRE1. However, lipid saturation can activate IRE1 through a mechanism that appears independent of its luminal domain. It would be good to include some potential discussion of how lipid saturation can bypass this oligomerization step to promote IRE1 oligomerization and/or autophosphorylation, especially because the presented results suggest IRE1 would be monomeric in the absence of the luminal domain.

This is an interesting comment, and we actually pursued this direction briefly before submitting the initial version of the manuscript. What we found when we induced lipid stress by loading cells with palmitate was that diffusion in the ER membrane was severely slowed down, both for IRE1 molecules and control HaloTag constructs. A significant fraction of single molecules even appeared to be completely “stuck” in one place. While this is an intriguing observation, our method for detecting oligomerization is inextricably linked to measuring diffusive trajectories, so if diffusion in the ER is so massively perturbed by lipid stress, it is difficult for us to make meaningful mechanistic conclusions. Rather than presenting these confusing results to the readers, we added the following sentences to the discussion:

“… Additionally, it remains to be determined how other signals (such as lipid saturation, which has been found to activate IRE1 independently of its lumenal domain (REF Volmer et al., 2013)), impact IRE1’s oligomeric state and phosphorylation. Based on prior work in the field, it is likely that changes in the ER membrane’s lipid composition directly drive IRE1 molecules into phosphorylation-competent small oligomers via key features of its transmembrane helix (REF Kono et al., 2017).”

Reviewer #2 (Recommendations for the authors):Determining which form(s) of IRE1a are active and inactive in cells is fundamental to understanding the Unfolded Protein Response. A single particle tracking imaging approach is highly appropriate and probably the best approach. Unfortunately, some fundamental issues significantly dampen my enthusiasm for this manuscript.

We fully understand the reviewer’s concerns and believe that we have addressed them in the revised version of the manuscript, as detailed below.

The main issue with the manuscript is the lack of apparent clustering of the IRE1a-Halo reporter. Given all of the other studies that observe clustering with different IRE1a reporters with different fusions in different cell types, including multiple studies from the Walter lab, the lack of clustering raises the reasonable question of: What is being studied here? The IRE1a clusters and clustering behavior described in the recent Belyy et al., study would likely produce very different % correlated trajectory results. If the results in the current study are unique to the reporter, it's unclear what the relevance is to the rest of the ER stress field. To provide confidence in this system, I think the authors need to determine 1. why there is a discrepancy and 2. which system(s) accurately report on IRE1a behavior. I appreciate that these tasks represent a significant amount of work and I am not making these suggestions lightly.

As described in our response to the public review, we strongly believe that this apparent discrepancy in IRE1’s clustering behavior is rooted in its expression levels. First, the IRE1-HaloTag construct used in the present study is fully capable of assembling into stress-induced puncta when it is overexpressed (see new Figure 1 —figure supplement 3). Second, conventional fluorescent protein fusion reporters do not always form clusters, as demonstrated in a parallel manuscript by Gómez-Puerta et al., (https://doi.org/10.1101/2021.11.15.468613). Third, our earlier work with an exogenously expressed IRE1-mNeonGreen construct (Belyy et al., PNAS 2020) showed that nearly half of all cells overexpressing IRE1-mNeonGreen never form detectable clusters, despite IRE1 in these cells being fully active. We have clarified each of these points in the Discussion section, which now includes the following text:

“The substantially lower expression levels achieved in this study are likely the reason that we did not observe endogenously tagged IRE1 assembling into large clusters. It is clear that the lack of large cluster formation is not a defect of the HaloTag construct, since the same construct readily assembles into stress-induced puncta when overexpressed (Figure 1 —figure supplement 3). Conversely, we can rule out the possibility that IRE1 clustering is an artifact of fluorescent protein fusions. In a parallel study by Gómez-Puerta et al., the authors expressed IRE1-GFP in HeLa cells at near-endogenous levels and demonstrated that it remains fully active despite not forming microscopically detectable clusters (56). Furthermore, in an earlier study from our own lab, we found that nearly one half of cells overexpressing an IRE1-mNeonGreen reporter did not form clusters over the time course of stress, despite fully splicing XBP1 mRNA (22). We also do not contend that large IRE1 clusters artificially result from overexpression: IRE1 levels were found to be highly variable across a panel of cancer cell lines (57, 58) and it is reasonable to suspect that large clusters of endogenous IRE1 do form in cell lines with elevated amounts of IRE1 protein.”

Regarding the nature of the discrepancy, it likely revolves around the Halo Tag, the JF dyes, and/or the insertion site. Either the new reporter sets a new standard for an inert tag or the new reporter will raise a cautionary note for the Halo Tag system. One useful approach would be the addition of a minimal epitope tag to IRE1, such as an HA tag, and analysis of the reporter's behavior with immunofluorescence of stressed and unstressed cells. It is probably the most physiologic system, short of a great anti-IRE1a IF antibody. With such a reporter, please assess whether IRE1a-epitope tag clusters form in stressed and not in unstressed cells. If affirmed, then the authors will need to dissect and optimize the Halo Tag system. Note that an HA tag has been used in yeast and stress induced-Ire1 clustering was readily observed by Kimata et al. If no clusters are observed with stressed epitope-tagged mammalian IRE1a, it would be helpful for the ER stress field for the authors to determine how fluorescent proteins, even robust monomers such as mNG, impact clustering and whether a significant re-evaluation of IRE1a clustering models will be necessary.

This is closely related to our response to the previous point. Since IRE1-HaloTag labeled with JF dyes is fully capable of clustering when overexpressed while IRE1-GFP can be active without clustering, we do not see this discrepancy as an issue with the tag but rather as a question of expression levels. We believe that this point is also addressed in our revised Discussion section, as cited above. Additionally, we included the following text in the Results section:

“The lack of clustering was not a defect of the IRE1-HaloTag fusion construct, since overexpressing the same IRE1-HaloTag protein by transient transfection resulted in readily observed stress-induced clusters (Figure 1 —figure supplement 3).”

Reviewer #3 (Recommendations for the authors):Given that, previously, IRE1 structural rearrangements have been tied to the XBP1-splicing-versus-RIDD functionalities of IRE1, does the IRE1a-HALO activate RIDD to the same extent and with the same kinetics as endogenous?

We thank the reviewer for raising this point. In response, we carried out qPCR experiments on both IRE1-HaloTag clones and showed that they are capable of RIDD activity in addition to XBP1 splicing. Please see the new Figure 1 —figure supplement 2 for details.

The tracking of multimerization through movement raises two related questions. One is whether the automated visualization is then biased for movement, such that, if higher ordered structures of IRE1 were relatively immobile, they would be missed. The flip side of this coin is that, presumably, embedded within the already-existing data is information on whether ER stress changes the mobility of IRE1, comparing correlated tracks to non-correlated tracks, and comparing correlated tracks in the presence and absence of stress.

This is yet another great question. Inspired by the reviewer’s suggestions, we looked more carefully at our existing single-particle data and showed that ER stress does not induce significant changes in IRE1 diffusion (unlike lipid stress, which does so profoundly and which precluded us from using those data in this manuscript; please see our response to reviewer 1’s last question). The relevant data are shown in the new Figure 3 —figure supplement 4. The reviewer is correct that very large immobile structures would be missed by our approach. However, we see no evidence for the existence of such structures in our data either by visual inspection of microscopy images or by quantitative analysis of single-particle diffusive trajectories.